# Causal Feature Selection via Orthogonal Search

**Ashkan Soleymani**[*]                                                                                         *ashkanso@mit.edu*
*Department of Electrical Engineering and Computer Science*
*Massachusetts Institute of Technology, Cambridge, US*

**Anant Raj**[*]                                                                                                    *anant.raj@inria.fr*
*Inria, Ecole Normale Supérieure*
*PSL Research University, Paris, France*

**Stefan Bauer**                                                                                                      *baue@kth.se*
*KTH, Stockholm, Sweden*

**Bernhard Schölkopf**                                                                                    *bs@tuebingen.mpg.de*
*Max Planck Institute for Intelligent Systems*
*Tübingen, Germany*

**Michel Besserve**                                                                          *michel.besserve@tuebingen.mpg.de*
*Max Planck Institute for Intelligent Systems*
*Tübingen, Germany*

**Reviewed on OpenReview:** *https://openreview.net/forum?id=Q54jBjc896*

## Abstract

The problem of inferring the direct causal parents of a response variable among a large set of explanatory variables is of high practical importance in many disciplines. However, established approaches often scale at least exponentially with the number of explanatory variables, are difficult to extend to nonlinear relationships and are difficult to extend to cyclic data. Inspired by *Debiased* machine learning methods, we study a one-vs.-the-rest feature selection approach to discover the direct causal parent of the response. We propose an algorithm that works for purely observational data while also offering theoretical guarantees, including the case of partially nonlinear relationships possibly under the presence of cycles. As it requires only one estimation for each variable, our approach is applicable even to large graphs. We demonstrate significant improvements compared to established approaches.

## 1 Introduction

Identifying causal relationships is a profound and hard problem pervading experimental sciences such as biology (Sachs et al., 2005), medicine (Castro et al., 2020), earth system sciences (Runge et al., 2019), or robotics (Ahmed et al., 2020). While randomized controlled interventional studies are considered the gold standard, they are in many cases ruled out by financial or ethical concerns (Pearl, 2009; Spirtes et al., 2000). In order to improve the understanding of a system and help design relevant interventions, the subset of causes that have a direct effect (*direct causes/direct causal parents*) often needs to be identified based on observations only. This paper assumes a structural equation model (SEM) comprising (1) a set of $d$ covariates represented by random vector $X \in \mathbb{R}^d$ whose values are determined by a uniquely solvable set of $d$ structural equations, possibly non-linear and possibly including cycles and confounding (2) a response variable $Y \in \mathbb{R}$, who is not a parent of any $X$ and whose value is determined by a linear structural equation of the form,

$$Y := \langle \theta, X \rangle + U , \ \text{ with } \theta \in \mathbb{R}^d, \tag{1}$$

---

[*] Equal contributions

where $U$ is an exogenous variable with zero mean, independent from any other exogenous variables of the SEM and $\langle \cdot, \cdot \rangle$ denotes the inner product. Such a SEM is exemplified in Figure 1. Uniquely solvability of SEMs amounts to not having self-cycles in the causal structure, but any other arbitrary non-linear cyclic structure between covariates is allowed (Bongers et al., 2021), possibly including hidden confounders, as long as there is no hidden confounder for the response variable (this would violate the assumption of independence of $U$). Practically speaking, almost all causal discovery applications lie under the umbrella of simple SCMs (Bollen, 1989; Sanchez-Romero et al., 2019). Besides, the assumption of not having self-cycles is usually assumed not-limiting in the literature (Lacerda et al., 2012; Rothenhäusler et al., 2015; Bongers et al., 2016).

In this paper, we investigate how to find the direct causes of $Y$ among a high-dimensional vector of covariates $X$. From our formulation, a given entry of $\theta$ should be non-zero if and only if the variable corresponding to that particular coefficient is a direct causal parent (Peters et al., 2017), e.g., $X_1$ and $X_2$ in Figure 1. We restrict ourselves to the setting of *linear direct causal effects* of $Y$ (LDC, as specified in Equation 1) and *no feature descending from $Y$* (NFD). LDC is justified as an approximation when the effects of each causal feature are weak such that the possibly non-linear effects can be linearized; NFD is justified in some applications where we can exclude any influence of $Y$ on a covariate. This is, for example, the case when $X$ are genetic factors, and $Y$ is a particular trait/phenotype. Our method, in particular, comes handy in this case due to the relatively complex non-linear cyclic structure of these genetic factors in high-dimensional regimes (Yao et al., 2015; Meinshausen et al., 2016; Warrell & Gerstein, 2020).

While applicable to full graph discovery rather than the simplified problem of finding causal parents, state-of-the-art methods for causal discovery often rely on strong assumptions or the availability of interventional data or have prohibitive computational costs explained in section 1.1. In addition to and despite their strong assumptions, causal discovery methods may perform worse than simple regression baselines (Heinze-Deml et al., 2018; Janzing, 2019; Zheng et al., 2018). However, estimating $\theta$ in high dimensional settings (i.e. # observations $<< d$) using unregularized least squares regression, will lead to identifiability problems since there can be infinitely many possible choices for $\theta$ recovered with equivalent prediction accuracy for regressing $Y$ (Bühlmann & Van De Geer, 2011). On the other hand, when using a regularized method such as Lasso, a critical issue is the bias induced by regularization (Javanmard & Montanari, 2018). While regularized and unregularized Least Squares as well as debiased Lasso are applicable to the same problem, they have different statistical characteristics and we will explore their empirical performance in extensive experiments.

Double ML approaches (Chernozhukov et al., 2018a) have shown promising bias compensation results in the context of high dimensional observed confounding of a single variable. In the present paper, we use this approach to find direct causes among a large number of covariates. Our key contributions are:

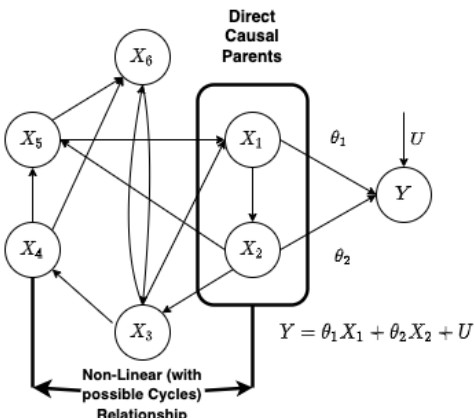

- We show that under the assumption that no feature of $X$ is a child of $Y$, the Double ML (Chernozhukov et al., 2018) principle can be applied in an iterative and parallel way to find the subset of direct causes with observational data.
- Our approach has a computational complexity requirement polynomial (fast) time in dimension $d$.
- Our method provides asymptotic guarantees that the set can be recovered from observational data. Importantly, this result neither requires linear interactions among the covariates, faithfulness, nor acyclic structure.
- Extensive experimental results demonstrate the state-of-the-art performance of our method. Our approach significantly outperforms all other methods (even though underlying data generation conditions favor them), especially in the case of non-linear interactions between covariates, despite relying only on linear projection.

Figure 1: Graphical representation of Causal Feature Selection in our setting, for the case of two direct causal parents of $Y$, $X_1$ and $X_2$, out of variables $\{X_1, \cdots, X_6\}$, such that $Y = \theta_1 X_1 + \theta_2 X_2 + U$, $U$ being an independent zero-mean noise. We propose an approach to find $X_1$ and $X_2$ under assumptions discussed in the text. An example of this setup in the real-world is finding genes which directly cause a phenotype.

## 1.1 Related work

The question of finding direct causal parents is also addressed in the
literature as mediation analysis (Baron & Kenny, 1986; Hayes, 2017; Shrout & Bolger, 2002). Several principled approaches have been proposed (relying, for instance, on Instrumental Variables (IVs)) (Angrist & Imbens, 1995; Angrist et al., 1996; Bowden & Turkington, 1990) to test for a single direct effect in the context of specific causal graphs. Extensions of the IV-based approach to generalized IVs-based approaches (Brito & Pearl, 2012; Van der Zander & Liskiewicz, 2016) are the closest known result to discovering direct causal parents. However, no algorithm is provided in Brito & Pearl (2012) to identify the instrumental set. Subsequently, an algorithm is provided in Van der Zander & Liskiewicz (2016) for discovering the instrumental set in the simple setting where all the interactions are linear and the graph is acyclic. In contrast, our method allows non-linear cyclic interaction amongst the variables.

Several other works have also tried to address the problem of discovering causal features. The authors review work on causal feature selection in Guyon & Aliferis (2007). More recent papers on causal feature selection have appeared since (Cawley, 2008; Paul, 2017; Yu et al., 2018), but none of those claims to recover all the direct causal parents asymptotically or non-asymptotically as we do in our case. There has been another line of works on inferring causal relationships from observational data based on conditional independences, such as the PC-algorithm, which can be used for more general causal inference purposes, at the expense of extra assumptions, such as faithfulness, allowing inference of the Markov equivalence class through testing iterative testing of d-separation statements (Mastakouri et al., 2019; Pearl, 2009; Spirtes et al., 2000). In contrast, under our LDC, NFD and totally independent U assumptions, only a small subset of d-separation relationships are relevant, and further, these assumptions imply that the observed relevant conditional independences and associated d-separation are equivalent, so that a full set of faithfulness assumptions about conditioning sets besides $X_{-j}$ are not necessary.

Another approach is to restrict the class of interactions among the covariates and the functional form of the signal-noise mixing (typically considered additive) or the distribution (e.g., non-Gaussianity) to achieve identifiability (see (Hoyer et al., 2009; Peters et al., 2014)); this includes linear approaches like LiNGAM (Shimizu et al., 2006) and nonlinear generalizations with additive noise (Peters et al., 2011). For a recent review of the empirical performance of structure learning algorithms and a detailed description of causal discovery methods, we refer to (Heinze-Deml et al., 2018). Recently, there have been several attempts at solving the problem of causal inference by exploiting the invariance of a prediction under a causal model given different experimental settings (Ghassami et al., 2017; Peters et al., 2016). The computational cost to run both algorithms is exponential in the number of variables when aiming to discover the full causal graph.

Our method mainly takes inspiration from Debiased/Double ML method (Chernozhukov et al., 2018a) which utilizes the concept of orthogonalization to overcome the bias introduced due to regularization. We will discuss this in detail in the next section. Considering a specific example, the Lasso suffers from the fact that the estimated coefficients are shrunk towards zero, which is undesirable (Tibshirani & Wasserman, 2017). To overcome this limitation, a debiasing approach was proposed for the Lasso in several papers (Javanmard & Montanari, 2014; 2018; Zhang & Zhang, 2014). However, unlike our approach, Debiased Lasso methods do not recover all the non-zero coefficients of the parameter vector $\theta$ under the generic assumptions of the present work. To be more specific, (Javanmard & Montanari, 2018) is built upon the Equation (1) with the following differences to our setting: Noise $U$ is Gaussian; $X$ has independent zero-mean Gaussian rows with covariance matrix $\Sigma$ satisfying specific bounding conditions; sparsity conditions $s_0 = o(n/(\log p)^2)$[1] and $\min(s_\Omega, s_0) = o(\sqrt{n}/\log p)$ where $s_0$ and $s_\Omega$ are sparsity levels of the true coefficients $\theta$ and the precision matrix $\Omega = \Sigma^{-1}$ of $X$ respectively.

## 2 Methodology

Before describing the proposed method, we discuss our general strategy as well as Double ML and Neyman orthogonality in the next sections, which will be helpful in building the theoretical framework for our method.

---

[1] $p = d - 1$ is the covariate dimension.

## 2.1 Reduction to a nonparametric estimation problem

According to Equation (1), determining whether $X_j$ is a parent of $Y$ in our setting amounts to testing whether $\theta_j \neq 0$. Let $X_{-j} = X \setminus X_j$, this can be reduced to testing whether the following estimand vanishes:

$$\chi_j \triangleq \mathbb{E}\left[(Y - \mathbb{E}(Y \mid X_{-j}))(X_j - \mathbb{E}(X_j \mid X_{-j}))\right] \tag{2}$$

Indeed, $U$ independent of $X$ entails $Y - \mathbb{E}(Y \mid X_{-j}) = \theta_j(X_j - \mathbb{E}(X_j \mid X_{-j})) + U$. This leads to

$$\chi_j = \theta_j \mathbb{E}\left[(X_j - \mathbb{E}(X_j \mid X_{-j}))^2\right] = \theta_j \mathbb{E}\left[X_j(X_j - \mathbb{E}(X_j \mid X_{-j}))\right]. \tag{3}$$

Under mild assumptions, testing whether $\theta_j \neq 0$ thus reduces to testing whether $\chi_j \neq 0$. Equation (2) shows that $\chi_j$ constitutes a *non-parametric estimand*, i.e. a model-free functional of the observed data distribution. Nonparametric estimation results (Robins et al., 2008; Van der Laan et al., 2011; Chernozhukov et al., 2018a) make use of the *efficient influence function* of such estimand (see e.g. Hines et al. (2022)) to derive valid estimates and confidence bounds, while allowing the use of data adaptive estimation strategies, such as machine learning algorithms. The resulting strategies are known as *target learning* and *debiased/double machine learning*, and are suitable in challenging settings such as ours when $X$ is high dimensional with possibly non-linear dependencies among components.

## 2.2 Double Machine Learning (Double ML)

Double ML constitutes one possible way to derive efficient nonparametric estimates. We introduce it with the partial linear regression setting introduced in Chernozhukov et al. (2018a, Example 1.1). Given a fixed set of policy variables $D$ and control variables $X$ acting as common causes of $D$ and $Y$, we consider the partial regression model of Equation (4),

$$\begin{aligned} Y &= D\theta_0 + g_0(X) + U, \quad \mathbb{E}\left[U|X, D\right] = 0 \\ D &= m_0(X) + V, \quad \mathbb{E}\left[V|X\right] = 0, \end{aligned} \tag{4}$$

where $Y$ is the outcome variable, $U, V$ are disturbances and $g_0, m_0 : \mathbb{R}^d \to \mathbb{R}$ are (possibly non-linear) measurable functions. An unbiased estimator of the causal effect parameter $\theta_0$ can be obtained via the orthogonalization approach as in Chernozhukov et al. (2018a), which is obtained via the use of the "Neyman Orthogonality Condition" described below.

**Neyman Orthogonality Condition:** Let $W$ denote the collection of all observed variables. The traditional estimator of $\theta_0$ in Equation (4) can be simply obtained by finding the zero of the empirical average of a score function $\phi$ such that $\phi(W; \theta, g) = D^\top(Y - D\theta - g(X))$. However, the estimation of $\theta_0$ is sensitive to the bias in the estimation of the function $g$. Neyman (Neyman, 1979) proposed an orthogonalization approach to get an estimate for $\theta_0$ that is more robust to the bias in the estimation of nuisance parameter $(m_0, g_0)$. Assume for a moment that the true nuisance parameter is $\eta_0$ (which represents $m_0$ and $g_0$ in Equation (4)) then the orthogonalized "score" function $\psi$ should satisfy the property that the Gateaux derivative operator with respect to $\eta$ vanishes when evaluated at the true parameter values:

$$\partial_\eta \mathbb{E}\psi(W; \theta_0, \eta_0)[\eta - \eta_0] = 0. \tag{5}$$

One way to build such a score, following Chernozhukov et al. (2018a) [eq. (2.7)], is to start from a biased score associated to maximum likelihood-like estimate. Let $\ell(W; (\theta, \boldsymbol{\eta}))$ be the *log likelihood* function or another smooth objective for which the true parameter is the unique maximizer. The true parameter then satisfies $\mathbb{E}\partial_\theta \ell(W; (\theta_0, \boldsymbol{\eta_0})) = 0$, suggesting to start with $\partial_\theta \ell(W; (\theta_0, \boldsymbol{\eta_0}))$ as a (biased) score. In order to compensate the bias due to the nuisance parameters, we then subtract a linear function of the derivative of the likelihood with respect it, leading to the orthogonalized score

$$\psi(W; \theta, \boldsymbol{\eta}) = \partial_\theta \ell(W; (\theta, \boldsymbol{\eta})) - \boldsymbol{\mu}\partial_\eta \ell(W; (\theta, \boldsymbol{\eta})).$$

where $\boldsymbol{\mu}$ is determined by the constraint of Equation (5) (see proof of Proposition 6 in appendix). The corresponding Orthogonalized or Double/Debiased ML estimator $\breve{\theta}_0$ solves a constraint of vanishing empirical average of the

orthogonalized score, based on $n$-iid samples $\{W_i\}_{i=1..n}$ of the observed variables.

$$\frac{1}{n}\sum_{i=1}^{n}\psi(W_i;\check{\theta}_0,\hat{\eta}_0) = 0, \tag{6}$$

where $\hat{\eta}_0$ is the estimator of $\eta_0$ and $\psi$ satisfies condition in Equation (5). For the partially linear model discussed in Equation (4), the orthogonalized score function $\psi$ is,

$$\psi(W;\theta,\eta) = (Y - D\theta - g(X))(D - m(X)), \tag{7}$$

with $\eta = (m, g)$. This leads to an debiased estimator satisfying

$$\check{\theta}_0\frac{1}{n}\sum_i D_i(D_i - \check{m}_0(X_i)) = \frac{1}{n}\sum_i(Y_i - \check{g}_0(X_i))(D_i - \check{m}_0(X_i)). \tag{8}$$

which relies on the "double" use of machine learning algorithm: once to learn $\check{g}_0(X_i)$ and once to learn $\check{m}_0(X_i)$, hence the name *Double ML* for such estimator. We can further relate this approach to the design an estimator of the non-parametric estimand of previous section.

Indeed by subtracting $\check{\theta}_0\frac{1}{n}\sum_i \check{m}_0(X_i)(D_i - \check{m}_0(X_i))$ on both sides of eq. (8), we get

$$\check{\theta}_0\frac{1}{n}\sum_i(D_i - \check{m}_0(X_i))^2 = \frac{1}{n}\sum_i(Y_i - \check{\theta}_0\check{m}_0(X_i) - \check{g}_0(X_i))(D_i - \check{m}_0(X_i)). \tag{9}$$

Noticing that $\mathbb{E}[Y|X] = \theta_0\mathbb{E}[D|X] + g_0(X) = \theta_0 m_0(X) + g_0(X)$, the term $\check{\theta}_0\check{m}_0(X_i) + \check{g}_0(X_i)$ in eq. (9) appears as an ML estimator of $\mathbb{E}[Y|X]$, such that we recognize on the right hand side of Equation (9) a Double ML estimator of $\mathbb{E}[(Y - \mathbb{E}[Y|X])(D - \mathbb{E}[D|X])]$, which is a special case of the non-parametric estimand $\chi_j$ defined in Equation (3), for the setting $X_j = D$ and $X = X_{-j}$. In practice, we directly learn an ML estimator of $\mathbb{E}[Y|X]$ by predicting $Y$ using $X$, relying on the double robustness of the $\chi_j$ estimands (Smucler et al., 2019), as described in section 2.5.

**From Double ML to Causal Discovery:** The distinction between policy variables and confounding variables is not always known in advance. Fortunately, as described in section 2.1, Double ML relies on estimating a non-parameteric estimand that does only depend on observational data and not on the causal model. This will allow us to exploit the same approach iteratively in the setting of causal discovery. To this end, we consider a set of variables $X = \{X_1, X_2, \cdots X_d\}$ which includes direct causal parents of the outcome variable $Y$ as well as other variables. We also reiterate our assumption that the relationship between the outcome variable and direct causal parents of the outcome variable is linear. The relationship among other variables can be cyclic and nonlinear. We now provide a general approach to scanning putative direct causes scaling "polynomially" in their number (see *Computational Complexity* paragraph in next section), based on the application of a statistical test and Double ML estimators. We describe first the algorithm and then provide theoretical support for its performance.

## 2.3 Informal Search Algorithm Description

Pseudo-code for our proposed method (CORTH Features) is in Algorithm 1. The idea is to do a one-vs-rest split for each variable in turn and estimate the link between that particular variable and the outcome variable using Double ML. To do so, we decompose Equation (1) to single out a variable $D = X_k$ as policy variable and take the remaining variables $Z = X_{-k} = X \backslash X_k$ as multidimensional control variables, and run Double ML estimation assuming the partial regression model presented in Section 2.2, which now takes the form

$$\begin{aligned} Y &= D\theta_k + g_k(Z) + U, \;\; \mathbb{E}\left[U|Z, D\right] = 0, \\ D &= m_k(Z) + V, \;\; \mathbb{E}\left[V|Z\right] = 0. \end{aligned} \tag{10}$$

The step-wise description of our estimation algorithm goes as follows:

(a) Select one of the variables $X_i$ to estimate its (hypothetical) linear causal effect $\theta$ on $Y$.

(b) Set all of the other variables $X_{-i}$ as the set of possible confounders.

(c) Use the Double ML approach to estimate the parameter $\theta$ i.e. the causal effect of $X_i$ on $Y$.

(d) If the variable $X_i$ is not a causal parent, the distribution of the conditional covariance $\chi_i$ (Proposition 3) is a Gaussian centered around zero. We use a simple normality test for $\chi_i$ to select or discard $X_i$ as one of the direct causal parents of $Y$.

We iteratively repeat the procedure on each of the variables until completion. Pseudo-code for the entire procedure is given below in Algorithm 1. Guaranties for this approach to identify the true parents rely on the assumptions stated in Section 2.5, Equations ( 13-15). They notably allow for hidden confounders between covariates, as long as those are not direct causes of $Y$, not descendent of $Y$. On the contrary, if $Y$ is an ancestor of any covariate, the search algorithm may fail in both directions (false positive and false negative).

Note that Equation (10) is not necessarily a correct structural equation model to describe the true underlying causal structure. In general, for instance, when $D$ actually causes $Z$, it is non-trivial to show that the Double ML estimation of parameter $\theta_k$ will be unbiased (see Section 2.4).

---

**Algorithm 1** Efficient Causal Orthogonal Structure Search (CORTH Features)

---

1: **Input:** response $Y \in \mathbb{R}^N$, covariates $\mathbb{X} \in \mathbb{R}^{N \times d}$, significance level $\alpha$, number of partitions $K$.

2: Split $N$ observations into K random partitions, $I_k$ for $k = 1, \ldots, K$, each having $n = N/K$ samples.

3: **for** $i = 1, \ldots, d$ **do**

4:     **for** Subsample $k \in [K]$ **do**

5:         $D_k \leftarrow X_i^{[k]}$ and $Z_k \leftarrow X_{\setminus i}^{[k]}$

6:         Fit $m_i^{[\setminus k]}(Z_{\setminus k})$ to $D_{\setminus k}$ and fit $g_i^{[\setminus k]}(Z_{\setminus k})$ to $Y^{[\setminus k]}$

7:         $\hat{V}_{ij}^{[k]} \leftarrow D_{kj} - m_i^{[\setminus k]}(Z_{kj})$, for all $j \in I_k$

8:         $\breve{\theta}_i^{[k]} \leftarrow \left( \frac{1}{n} \sum_{j \in I_k} \hat{V}_{ij}^{[k]} D_{kj} \right)^{-1} \frac{1}{n} \sum_{j \in I_k} \hat{V}_{ij}^{[k]} (Y_{ij}^{[k]} - g_{ij}^{[\setminus k]}(Z_{kj}))$

9:         $\hat{\chi}_i^{[k]} \leftarrow \frac{1}{n} \sum_{j \in I_k} \left( - Y_j^{[k]} m_{ij}^{[\setminus k]}(Z_{kj}) - D_{kj} g_{ij}^{[\setminus k]}(Z_{kj}) + m_{ij}^{[\setminus k]}(Z_{kj}) g_{ij}^{[\setminus k]}(Z_{kj}) + Y_j^{[k]} D_{kj} \right)$

10:         $(\hat{\sigma}_i^{[k]})^2 \leftarrow \frac{1}{n} \sum_{j \in I_k} \left( - Y_j^{[k]} m_{ij}^{[\setminus k]}(Z_{kj}) - D_{kj} g_{ij}^{[\setminus k]}(Z_{kj}) + m_{ij}^{[\setminus k]}(Z_{kj}) g_{ij}^{[\setminus k]}(Z_{kj}) + Y_j^{[k]} D_{kj} - \hat{\chi}_i^{[k]} \right)^2$

11:     **end for**

12:     $\hat{\theta}_i \leftarrow \frac{1}{K} \sum_{k \in K} \breve{\theta}_i^{[k]}$, $\hat{\chi}_i \leftarrow \frac{1}{K} \sum_{k \in K} \hat{\chi}_i^{[k]}$ and $\hat{\sigma}_i^2 \leftarrow \frac{1}{K} \sum_{k \in K} (\hat{\sigma}_i^{[k]})^2$

13: **end for**

14: **for** $i \in [d]$ **do**

15:     Gaussian normality test for $\hat{\chi}_i \approx N\left( 0, \frac{\hat{\sigma}_i^2}{N} \right)$ with $\alpha$ significance level and select $i^{\text{th}}$ feature if null-hypothese is rejected.

16: **end for**

17: **Return** Decision Vector

---

**Remarks on Algorithm 1:** $X_i^{[k]}$ is a vector which corresponds to the samples chosen in the $k^{th}$ subsampling procedure, $X_{\setminus i}^{[k]} = (X_1^{[k]}, \ldots, X_{i-1}^{[k]}, X_{i+1}^{[k]}, \ldots, X_d^{[k]})$ for any $i \in [d]$. In general the subscript $i$ represents the estimation for the $i^{th}$ variable and super-script $k$ represents the $k^{th}$ subsampling procedure. $K$ represents the set obtained after sample splitting. $m_i^{[\setminus k]}$ are (possibly nonlinear) parametric functions fitted using $(1^{st}, \ldots, k-1^{th}, k+1^{th}, \ldots, K^{th})$ subsamples.

**Computational Complexity:** For each subset randomly selected from the data, we fit two lasso estimators. Accelerated coordinate descent (Nesterov, 2012) can be applied to optimize the lasso objective. To achieve $\varepsilon$ error, $\mathcal{O}\left( d \sqrt{\kappa_{\max}} \log \frac{1}{\varepsilon} \right)$ number of iterations are required where $\kappa_{\max}$ is the maximum of the two condition number for both the problems and each iteration requires $\mathcal{O}(nd)$ computation. Hence, the computational complexity of running our approach is only polynomial in $d$.

### 2.4 Orthogonal Scores

Now we describe the execution of our algorithm for a simple graph with 3 nodes. Let us consider the following linear structural equation model as an example of our general formulation:

$$Y := \theta_1 X_1 + \theta_2 X_2 + \varepsilon_3, \ X_2 := a_{12} X_1 + \varepsilon_2, \text{ and } X_1 := \varepsilon_1. \tag{11}$$

**Example 1.** *Consider the system of structural equation given in Equation (10). If $\varepsilon_1$, $\varepsilon_2$ and $\varepsilon_3$ are independent uncorrelated noise terms with zero mean, Algorithm 1 will recover the coefficients $\theta_1$ and $\theta_2$.*

A detailed proof is given in Appendix A.1. While the estimation of the parameter $\theta_1$ is in line with the assumed partial regression model of Equation (11), the estimation of $\theta_2$ does not follow the same. However, it can be seen from the proof that $\theta_2$ can also be estimated from the orthogonal score in Equation (7).

We now show that this result holds for a more general graph structure given in Figure 2, allowing for non-linear cyclic interactions among features.

**Proposition 2.** *Assume the structural causal model of Figure 2, with (possibly non-linear and confounded) assignments between elements of $X = [X_k, X_{-k}^\top]^\top$, with $X_{-k} = [Z_1^\top, Z_2^\top]^\top$, parameterized by $\gamma = (\gamma_1, \gamma_2, \gamma_{12})$. Assume the unconfounded linear structural assignment $Y := X_k \theta + X_{-k}^\top \beta + U$, with $U$ zero mean random variable with finite variance $\sigma_U^2 > 0$, independent of $X$. Then, the score*

$$\psi(W; \theta, \beta) = (Y - X_k \theta - X_{-k}^\top \beta)(X_k - r_{XX_{-k}} X_{-k}), \tag{12}$$

*with $r_{XX_{-k}} = \mathbb{E}[X_k X_{-k}^\top] \mathbb{E}[X_{-k} X_{-k}^\top]^{-1}$, follows the Neyman orthogonality condition for the estimation of $\theta$ with nuisance parameters $\eta = (\beta, \gamma)$ which reads*

$$\mathbb{E}\left[(Y - X_k \theta - X_{-k}^\top \beta)(X_k - r_{XX_{-k}} X_{-k})\right] = 0.$$

Please refer to Appendix A.2 for the proof. Applying Equation (6), this leads to the debiased estimator

$$\check{\theta} = \frac{\sum_i (Y_i - X_{-ki}^\top \check{\beta})(X_{ki} - \check{r}_{XX_{-k}} X_{-k})}{\sum_i X_{ki}(X_{ki} - \check{r}_{XX_{-k}} X_{-ki})}.$$

which relies on ML estimates $\check{\beta}$ and $\check{r}_{XX_{-k}}$. Comparing the score in Equation (21) with the score in Equation (7), there are two takeaways from Proposition 2: (i) the orthogonality condition remains invariant irrespective of the causal direction between $X_k$ and $Z$, and (ii) the second term in Section 2.4 replaces function $m$ by the (unbiased) linear regression estimator for modelling all the relations; given that the relation between $Z$ and $Y$ is linear, even if relationships between $Z$ and $X_k$ are non-linear (See Appendix B for concrete examples). Combining with the Double ML theoretical results (Chernozhukov et al., 2018a), this suggests that regularized predictors based on Lasso or ridge regression are tools of choice for fitting functions $(m, g)$.

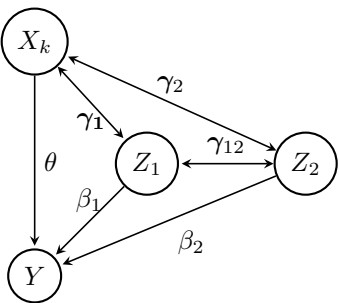

Figure 2: A generic example of identification of a causal effect $\theta$ in the presence of causal and anti-causal interactions between the causal predictor and other putative parents, and possibly arbitrary cyclic and nonlinear assignments for all nodes except $Y$ (see Proposition 2). We have $X_{-k} = Z_1 \cup Z_2$.

### 2.5 Statistical Test

We now provide a theoretically grounded statistical decision criterion for the direct causes after the model has been fitted. Consider $(Y, X)$, $Y \in \mathbb{R}$, $X \in \mathbb{R}^d$, satisfying

$$Y = \langle \theta, X \rangle + U, \tag{13}$$

$$\mathbb{E}(Y^2) < \infty, \ \mathbb{E}(U^2) < \infty, \ \mathbb{E}(U) = 0, \mathbb{E}(U \mid X) = 0, \text{ and } \mathbb{E}(\|X\|_2^2) < \infty, \tag{14}$$

$$\mathbb{E}\left[(X_j - \mathbb{E}(X_j \mid X_{-j}))^2\right] \neq 0, \quad \text{for all } j \in \{1, \dots, d\}, \tag{15}$$

where $U$ is an exogenous variable and $X_{-j}$ represents all the variables except $X_j$. The assumptions made with the above formulation are standard in the orthogonal machine learning literature (Rotnitzky et al., 2019; Smucler et al., 2019; Chernozhukov et al., 2018). They allow identifying causal parents based on estimates of conditional covariances $\chi_j$ defined in Equation (3)

**Proposition 3.** *Let* $PA_Y = \{j \in \{1, \ldots, d\} : \theta_j \neq 0\}$. *Then under the conditions given in Equations* (13) *to* (15), *for each* $j \in \{1, \ldots, p\}$

a) $\chi_j = \theta_j \mathbb{E}\left[(X_j - \mathbb{E}(X_j \mid X_{-j}))^2\right]$ *and* $j \in PA_Y$ *if and only if* $\chi_j \neq 0$.

b) *We also have (with notations of Prop. 2)* $\chi_j = \mathbb{E}\left[(Y - \mathbb{E}(Y \mid X_{-j}))\left(X_j - r_{XX_{-j}}X_{-j}\right)\right]$.

The proof is given in appendix A.3. There are two main implications of the results provided in Proposition 3. (i) $\chi_j$ is non-zero only for direct causal parents of the outcome variable, and $\chi_j$ has double robustness property as shown in (Rotnitzky et al., 2019; Smucler et al., 2019; Chernozhukov et al., 2018). Having double robustness property means that while computing the empirical version of the $\chi_j$ which we denote as $\hat{\chi}_j$, one can use regularized methods like ridge regression or Lasso to estimate the conditional expectation (function $m$). Afterward, one can perform statistical tests on top of it to decide between zero or non-zero tests. (ii) In line with the above orthogonal score results, we see that this quantity can be estimated using linear (unbiased) regression to fit the function $m$, although interactions between features may be non-linear. Next, we discuss the variance of our estimator so that a statistical test can be used to identify causal parents. For the sake of convenience, the case of 2 partitions $(K = 2)^2$ is explained here.

**Variance of Empirical Estimates of** $\chi_j$**:** Suppose we have $n$ i.i.d. observations indicated by $\mathcal{D}_n = \{(X_i, Y_i), i = 1\ldots, n\}$. Randomly split the data in two halves, say $\mathcal{D}_{n1}$ and $\mathcal{D}_{n2}$. Take $j \in \{1, \ldots, d\}$. For $k = 1$ let $\overline{k} = 2$, for $k = 2$ let $\overline{k} = 1$. For $k = 1, 2$, compute estimates of $\widehat{\mathbb{E}^k}(Y \mid X_{-j})$ and $\widehat{\mathbb{E}^k}(X_j \mid X_{-j})$ using the data in sample $\overline{k}$. Following Smucler et al. (2019), we can use estimates of $\widehat{\mathbb{E}^k}(Y \mid X_{-j})$ and $\widehat{\mathbb{E}^k}(X_j \mid X_{-j})$ that are solutions of $\ell_1$-regularized regression problems to obtain square root N convergence guaranties. We use Lasso as the estimator for conditional expectation in the experiments. Now, we compute the cross-fitted empirical estimates of $\chi_j$ and associated empirical variances

$$\hat{\chi}_j^k = \mathbb{P}_{nk}\left[-Y\widehat{\mathbb{E}^k}(X_j \mid X_{-j}) - X_j\widehat{\mathbb{E}^k}(Y \mid X_{-j}) + \widehat{\mathbb{E}^k}(Y \mid X_{-j})\widehat{\mathbb{E}^k}(X_j \mid X_{-j}) + YX_j\right] \tag{16}$$

$$\left(\hat{\sigma}_j^k\right)^2 = \mathbb{P}_{nk}\left[\left(-Y\widehat{\mathbb{E}^k}(X_j \mid X_{-j}) - X_j\widehat{\mathbb{E}^k}(Y \mid X_{-j}) + \widehat{\mathbb{E}^k}(Y \mid X_{-j})\widehat{\mathbb{E}^k}(X_j \mid X_{-j}) + YX_j - \hat{\chi}_j^k\right)^2\right], \tag{17}$$

where $\mathbb{P}_{nk}$ denotes the empirical average over the $k$ half. Finally, let

$$\hat{\chi}_j = \frac{1}{2}\left(\hat{\chi}_j^1 + \hat{\chi}_j^2\right), \quad \hat{\sigma}_j^2 = \frac{1}{2}\left(\left(\hat{\sigma}_j^1\right)^2 + \left(\hat{\sigma}_j^2\right)^2\right). \tag{18}$$

Consistency of such estimators notably relies on sparsity assumptions for ground truth models in the asymptotic high-dimensional setting where covariate dimension $p(= d - 1)$ and sparsity is allowed to vary with number of samples $n$. In the notations below, we drop this dependency and specialize to our case.

**Definition 4** (Approximate Linear-Sparse class (ALS))**.** *Ground truth predictor* $c(X_{-j})$ *belongs to the approximately sparse class whenever there exists* $\theta^* \in \mathbb{R}^p$ *and a function* $r(X_{-j})$ *satisfying*

$$c(Z) = \langle\theta^*, \phi(Z)\rangle + r(Z), \text{ where } \|\theta^*\|_0 \leq s \text{ and } \mathbb{E}[r(Z)^2] \leq K(s\log(p)/n).$$

Theorem 1 of (Smucler et al., 2019) provides general conditions under which (see also (Chernozhukov et al., 2018)), when the estimators $\widehat{\mathbb{E}^k}(Y \mid X_{-j})$ and $\widehat{\mathbb{E}^k}(X_j \mid X_{-j})$ are Lasso-type regularized linear regressions.

**Proposition 5.** *Given eqs.* (13-15) *and* $j \in \{1, \ldots, d\}$. *Assume: (i) For true parameter in* (13), $\|\theta\|_0 \leq s$ *with* $s\log(p)/n \to 0$; *(ii)* $r_{XX_{-j}}X_{-j}$ *is in the ALS class with* $s\log(p)/n \to 0$; *(iii) All* $X_k$ *have support bounded by the same constant; (iv)* $U$ *is independent of* $X$ *and has tail decaying at least as fast as an exponential random variable;*

---

[2]Extension to arbitrary number of data partitions $(K \geq 2)$ is straightforward. Check Algorithm 1.

*(v) $\mathbb{E}[X_{-j}X_{-j}^\top]$ has eigenvalues lower and upper bounded; (vi) $\mathbb{E}[Y - \mathbb{E}[Y|X_{-j}]|X_{-j}]$ and $\mathbb{E}[X_j - \mathbb{E}[X_j|X_{-j}]|X_{-j}]$ are bounded with variance lower bounded.*
*Then the pair of estimators $(\widehat{\chi}_j, \widehat{\sigma}_j^2)$ of eq. (18), using $l_1$ regularization coefficient $\lambda \approx \sqrt{\log(p)/n}$ for both Lasso estimates, satisfies $\widehat{\chi}_j \overset{D}{\to} \mathcal{N}\left(\chi_j, \frac{\widehat{\sigma}_j^2}{n}\right)$ ., as $n \to +\infty$.*

All stated bounds are uniform across $n$ and strictly positive. Proof is provided in Appendix A.4 with the broader context on the doubly robust estimation framework. The provided conditions correspond to a concise special case of the general statements provided in Smucler et al. (2019). It is possible to relax some of these assumptions, notably to allow some misspecifications of the sparsity assumptions for the Lasso estimates. Overall, the mildness of these assumptions may explain the empirical success of this approach in the following section.

Under conditions of proposition 5, the test that rejects $\chi_j = 0$ when $|\widehat{\chi}_j| \geq 1.96\frac{\widehat{\sigma}_j}{\sqrt{n}}$ will have approximately 95% confidence level. The probability of rejecting the null when it is false is

$$P\left(|\widehat{\chi}_j| \geq 1.96\frac{\widehat{\sigma}_j}{\sqrt{n}}\right) \geq P\left(|\widehat{\chi}_j - \chi_j| \leq |\chi_j| - 1.96\frac{\widehat{\sigma}_j}{\sqrt{n}}\right) \to 1.$$

In order to account for multiple testing, we use Bonferroni correction.

**Conditional Independence Tests:** Asymptotically, the conditional independence testing between $Y$ and $X_j$ given $X_{-j}$ is also a possible solution for our proposed approach. Indeed, d-separation rules imply that true causes are conditionally dependent according to this test, while non-causes are conditionally independent (because $X_{-j}$ is not a collider under our NFD assumption). However, conditional independence testing is challenging in high-dimensional/non-linear settings. Kernel-based conditional independence testing is computationally expensive (Zhang et al., 2012). We used $\chi_j$ in the paper because it was already known from previous works (Smucler et al., 2019; Chernozhukov et al., 2018b) that it has double robustness property, which means one can use regularized methods like Lasso to estimate empirical conditional expectation from a finite number of samples and the empirical estimator is still unbiased with controlled variance. Our work is related to the recent work of (Shah & Peters, 2020), which proposes a conditional independence test whose proofs rely heavily on (Chernozhukov et al., 2018a). In this paper, we use for the first time such double ML-based tests for the search problem.

## 3 Experiments

### 3.1 Experimental Setup

To showcase performance of our algorithm, we conducted two sets of experiments: i) Comparison with causal structure learning methods (Casual and Markov Blanket discovery) using data consisted of DAGs with high number of observations-to-number of variables ratio ($n \gg d$) which is applicable to causal structure learning methods. Markov Blanket discovery methods are included since under NFD, faithfulness, and no-hidden-confounders assumptions, Markov Blanket of the target variable corresponds to the direct parents. Note that, faithfulness and no-hidden-confounders assumptions are not necessary for our method. These experiments are discussed in details in Section 3.1.1 ii) Comparison with inference by regression methods using data consisted of DAGs with high number of observations-to-number of variables ratio ($n \approx d$ and $n \ll d$) to illustrate performance in high-dimensional regimes. This part is explained thoroughly in Section 3.1.2

### 3.1.1 Causal Structure Learning

For every combination of number of nodes (#nodes), connectivity ($p_s$), noise level ($\sigma^2$), number of observations ($n$), and non-linear probability ($p_n$) (see Table C.1), 100 examples (DAGs) are generated and stored as csv files (altogether 72.000 DAGs are simulated, comprising a dataset of overall >10GB). For each DAG, $n$ samples are generated. We provide more details about the parameters (#nodes, $p_s$, $p_n$ and $n$) and data generation process in Appendix C.1.1. For future benchmarking, the generated files with the code will be made available later.

The baselines we compare our method against are categorized in two groups which are suitable for observational data: i) Causal Structure Learning methods: LiNGAM (Shimizu et al., 2006), order - independent PC (Colombo

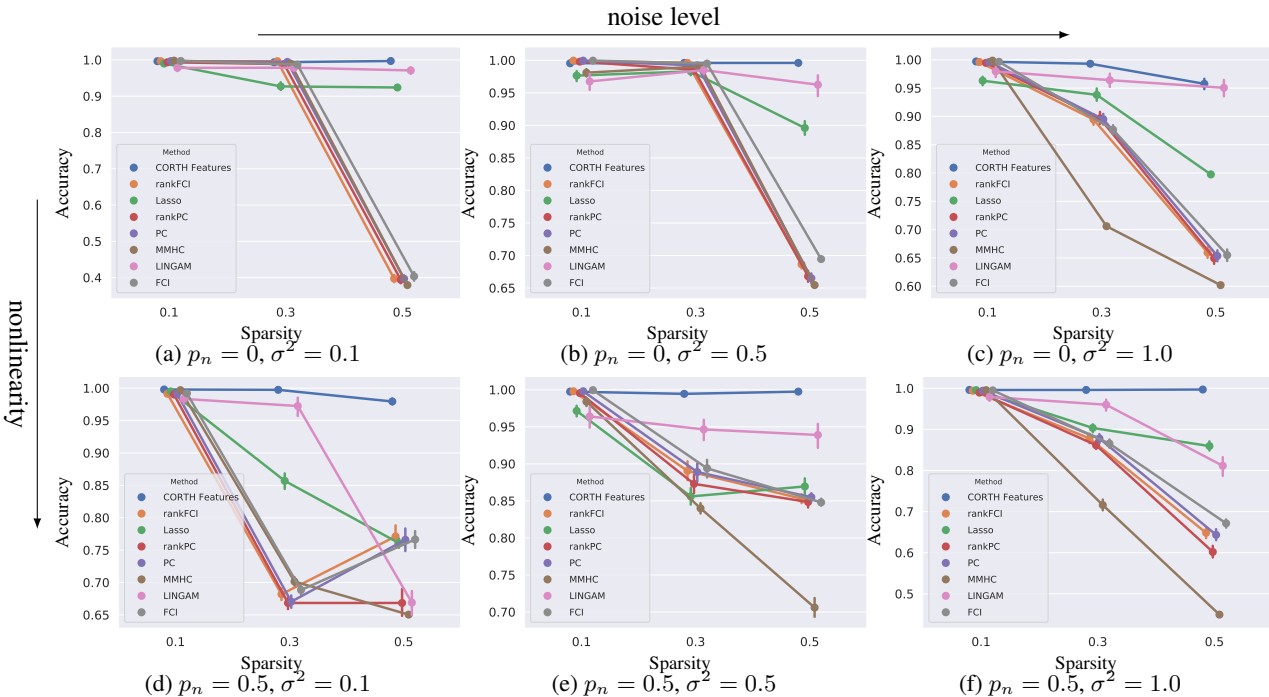

Figure 3: Overall performance for a single random DAG with 100 simulations for each setting, having 20 nodes and 500 observations.

& Maathuis, 2014), rankPC, MMHC (Tsamardinos et al., 2006), GES (Chickering, 2003), rankGES, ARGES (adaptively restricted GES (Nandy et al., 2016)), rankARGES, FCI+ (Claassen et al., 2013), PCI (Shah & Peters, 2020) and Lasso[3] (Tibshirani, 1996). ii) Markov Blanket discovery methods: Grow-Shrink (GS (Margaritis & Thrun, 1999)), Incremental Association Markov Blanket (IAMB (Tsamardinos et al., 2003b)), Max-Min Parents & Children (MMPC (Tsamardinos et al., 2003a)), FastIAMB (Yaramakala & Margaritis, 2005). and IAMB with FDR Correction (Pena, 2008). The "CompareCausalNetworks"[4] and "bnlearn: Bayesian Network Structure Learning, Parameter Learning and Inference"[5] R Packages are used to run most of the baselines methods. We use 10-fold cross-validation to choose the parameters of all approaches. As direction of the possible causes in the defined setting is determined, the non-directional edges inferred by some baselines, e.g., PC are evaluated as direct causes of the target variable.

### 3.1.2 Inference by Regression

Similar to the previous section, for every combination of parameters, 50 examples are generated and stored, which means 15000 DAGs overall. Details are provided in Appendix C.1.2 We compare our algorithm to methods for inference in regression models: Standard Regression, Lasso with exact post-selection inference (Lee et al., 2016), Debiased Lasso (Javanmard et al., 2015), Forward Stepwise Regression for active variables (Loftus & Taylor, 2014; Tibshirani et al., 2016), Forward Stepwise Regression for all variables (Loftus & Taylor, 2014; Tibshirani et al., 2016), LARS for active variables (Efron et al., 2004; Tibshirani et al., 2016), and LARS for all variables (Efron et al., 2004; Tibshirani et al., 2016). "selectiveInference: Tools for Post-Selection Inference" R Package [6] is leveraged to run most of these baselines. We used cross-validation to choose hyperparameters and confidence level for hypothesis testing considered is 90%.

**Regression Technique and Hyper-parameters:** We use Lasso as the estimator of conditional expectation for our method because the variance bound for $\chi_j$ with Lasso type estimator of conditional expectation is provided in equa-

---

[3]None-zero coefficients are reported.
[4]https://cran.r-project.org/web/packages/CompareCausalNetworks/index.html
[5]https://cran.r-project.org/web/packages/bnlearn/
[6]https://cran.r-project.org/web/packages/selectiveInference/

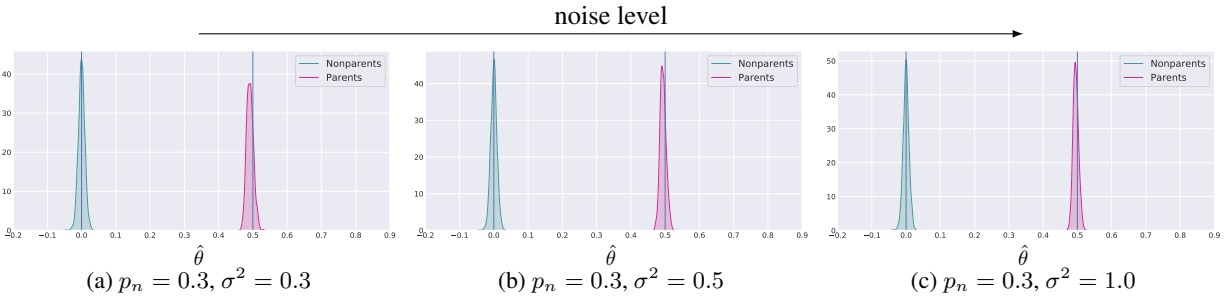

Figure 4: Distribution of the estimated $\theta$ values for the true and false causal parents in 100 simulations of the graph with 20 nodes, 20000 observations and 0.3 as connectivity. The vertical lines indicate the ground truth values for the linear coefficients corresponding to causal parents.

tion 17. Further, using more splits than 2 splits in the experiment relatively increases the performance of parameter estimation. See Figure 4 for parameter estimations.

**Evaluation:** Recall, Fall-out, Critical Success Index, Accuracy, F1 Score, and Matthews correlation coefficient (Matthews, 1975) are considered as metrics for the evaluation. These metrics are described in Appendix C.2.

### 3.2 Results

#### 3.2.1 Causal Structure Learning

Results aggregated by the number of observations (corresponding to 24000 simulations per entry in the table) are illustrated in Table 1[7]. Our method performs better than the competing baselines in terms of accuracy and F1 score, especially for more connected structures, despite data being generated according to DAG causal structures, which, dissimilar to our method, is an essential condition for them. To provide a visual comparison, we plot the accuracy of all methods w.r.t. the connectivity parameter ($p_s$) in Figure 3 for different values of $p_n$ and $\sigma^2$ on 1800 samples.

It can be observed that the accuracies of the competing baselines significantly drop with increasing noise level and nonlinearity, while our method is more robust to them. We also extensively compare all the metrics (Recall, Fall-out, Critical Success Index, Accuracy, F1 Score, and Matthews correlation coefficient) for all the methods in Appendix C.3.1. According to these metrics, our approach performs better than baselines in most cases regardless of the set of parameters used for generating data. Our method shows in particular stability in performance w.r.t. the number of nodes (Table C.3), partially non-linear relationships (Table C.4), connectivity (Table C.5), number of observations (Table C.7), and noise level (Table C.6). We also show the plot of parameter estimation for direct causal parents vs. non-causal parents in Figure 4. In the plots and tables, we denote our approach as `CORTH Features`.

#### 3.2.2 Inference by Regression

Analogous to previous part, results are aggregated by nonlinear probability (corresponding to 3750 simulations per entry in the table), number of observations (3000 simulations per entry in the table), connectivity (5000 simulations per entry in the table) and beta distribution parameters are provided in Tables C.8 to C.11. Based on these results, our method suggests more robustness w.r.t. the set of parameters used for generating data and relatively better performance compared to other methods.

### 3.3 Scaling Causal Inference to Large Graphs

Figure 5 shows the runtime of the method in secs as a function of the graph's size. Notice that the runtime of our algorithm in the log-log plot is roughly linear, supporting our above statement about the computational time being polynomial in $d$. As we used 5000 observations, additional overhead comes from cross-validation.

---

[7]Please refer to Appendix C.3.1 for thorough tables for all parameters.

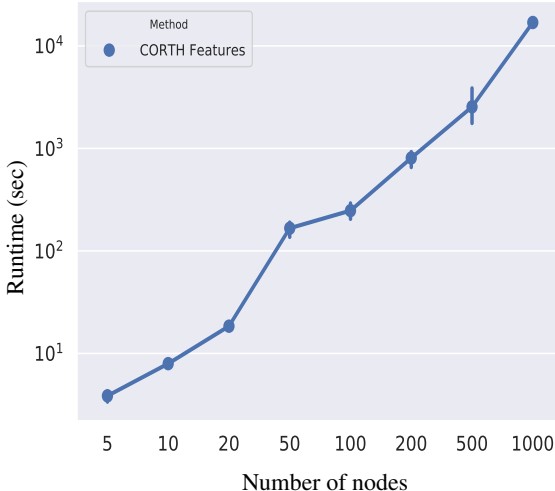

Figure 5: Runtime as a function of the number of variables for 10 simulations per number of nodes. In these simulations connectivity, number of observations, nonlinaer prob., and noise level are set to 0.3, 5000, 0, and 1 respectively.

| Method | Number of Observations | | | | | |
| | 100 | | 500 | | 1000 | |
| | ACC | F1 | ACC | F1 | ACC | F1 |
| --- | --- | --- | --- | --- | --- | --- |
| GES | 0.80 | 0.59 | 0.81 | 0.65 | 0.81 | 0.67 |
| rankGES | 0.79 | 0.56 | 0.81 | 0.64 | 0.81 | 0.65 |
| ARGES | 0.78 | 0.49 | 0.80 | 0.58 | 0.80 | 0.59 |
| rankARGES | 0.78 | 0.47 | 0.79 | 0.57 | 0.80 | 0.58 |
| FCI+ | 0.84 | 0.67 | 0.86 | 0.75 | 0.87 | 0.78 |
| LINGAM | 0.84 | 0.65 | 0.91 | 0.74 | 0.94 | 0.88 |
| PC | 0.83 | 0.64 | 0.86 | 0.73 | 0.87 | 0.75 |
| rankPC | 0.82 | 0.62 | 0.85 | 0.71 | 0.85 | 0.73 |
| MMPC | 0.77 | 0.37 | 0.82 | 0.53 | 0.83 | 0.57 |
| MMHC | 0.80 | 0.56 | 0.82 | 0.62 | 0.83 | 0.64 |
| GS | 0.79 | 0.43 | 0.84 | 0.59 | 0.86 | 0.62 |
| IAMB | 0.74 | 0.39 | 0.81 | 0.57 | 0.83 | 0.61 |
| Fast-IAMB | 0.80 | 0.46 | 0.84 | 0.59 | 0.86 | 0.62 |
| IAMB-FDR | 0.78 | 0.37 | 0.84 | 0.58 | 0.85 | 0.61 |
| PCI | 0.83 | 0.59 | 0.91 | 0.85 | 0.93 | 0.89 |
| Lasso | 0.87 | **0.81** | 0.89 | 0.85 | 0.89 | 0.85 |
| CORTH Features | **0.88** | 0.78 | **0.93** | **0.91** | **0.94** | **0.92** |

Table 1: Performance across all the settings for different number of observations (100, 500 and 1000). Each single entry in the table is averaged over 24000 simulations. Our method is almost state of the art in every case.

### 3.4 Real-World Data

We also apply our algorithm to a recent COVID-19 Dataset (Einstein, 2020) where the task is to predict COVID-19 cases (confirmed using RT-PCR) amongst suspected ones. For an existing and extensive analysis of the dataset with predictive methods, we refer to Schwab et al. (2020). We apply our algorithm to discover the features which directly cause the diagnosed infection. We found that the following were the most common causes across different runs of our approach: Patient age quantile, Arterial Lactic Acid, Promyelocytes, and Base excess venous blood gas analysis. Lacking medical ground truth, we report these not as corroboration of our approach but rather as a potential contribution to causal discovery in this challenging problem. It is encouraging that some of these variables are consistent with other studies Schwab et al. (2020). Details on data preprocessing and more results are available in Appendix D.

## 4  Discussion

A recent empirical evaluation of different causal discovery methods highlighted the desirability of more efficient search algorithms (Heinze-Deml et al., 2018). In the present work, we provide identifiability results for the set of direct causal parents, including the case of partially nonlinear cyclic models, as well as a highly efficient algorithm that scales well w.r.t. the number of variables and exhibits state-of-the-art performance across extensive experiments. Our approach builds on the Double ML method Chernozhukov et al. (2018a) and properties of conditional covariance estimands, which can leverage Lasso predictors to obtain consistent estimators in high dimensional settings Smucler et al. (2019). Theoretical properties of such approaches (Smucler et al., 2019, Section 2), such as *rate double robustness*, which accommodates non-strict (approximate) sparsity assumptions, as well as model double robustness, which allows some degree of model misspecification, may explain the empirically observed performance of our approach. Whilst not amounting to full causal graph discovery, identification of causal parents is of major interest in real-world applications, e.g., when assaying the causal influence of genes on the phenotype. A natural direction worth exploring is to extend this approach for discovering direct causal parents in the case when nonlinear relationships exist between the output variable and its direct causal parents.

### Acknowledgements

This work was supported by the German Federal Ministry of Education and Research (BMBF): Tübingen AI Center, FKZ: 01IS18039B .

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

# A  Causal Discovery via Orthogonalization

## A.1  Example 1

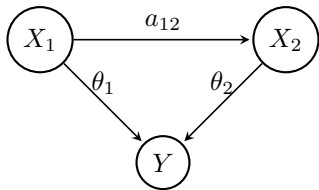

Figure A.1: An example with linear structural equations.

*Proof of Example 1* . Let us start from the easier case first (See Figure A.1) . Let us first try to estimate the coefficient of interaction between $X_2$ and $Y$ but it is also very clear that the estimation of $\theta_2$ will be unbiased as the given setting precisely match with the double machine learning setting. However, we will see in this example that given the population, $\theta_1$ can be approximated as well. Let us write down the structural equation model first:

$$
\begin{aligned}
Y &:= \theta_1 X_1 + \theta_2 X_2 + \varepsilon_3 \\
X_2 &:= a_{12} X_1 + \varepsilon_2 \\
X_1 &:= \varepsilon_1.
\end{aligned}
\tag{19}
$$

From the set of equations we have:

$$
X_1 = a_{12}^{-1} X_2 - a_{12}^{-1} \varepsilon_2.
$$

Let also denote $\mathbb{E}[\varepsilon_1^2] = \sigma_1^2$ and $\mathbb{E}[\varepsilon_2^2] = \sigma_2^2$. Hence, $\mathbb{E}[X_1^2] = \sigma_1^2$, $\mathbb{E}[X_1 X_2] = a_{12}\sigma_1^2$ and $\mathbb{E}[X_2^2] = a_{12}\mathbb{E}[X_1 X_2] + \mathbb{E}[\varepsilon_2 X_2] = a_{12}^2\sigma_1^2 + \sigma_2^2$.. Let us first try to find the regression co-efficient of fitting $X_2$ on $Y$.

$$
Y = \hat{\theta}_2 X_2 + \eta_1.
$$

Hence, $\hat{\theta}_2 = \frac{\mathbb{E}[X_2 Y]}{\mathbb{E}[X_2^2]}$ if $\eta$ is independent of $X_2$.

$$
\hat{\theta}_2 = \frac{\mathbb{E}[X_2 Y]}{\mathbb{E}[X_2^2]} = \frac{\mathbb{E}[X_2(\theta_1 X_1 + \theta_2 X_2 + \varepsilon_3)]}{\mathbb{E}[X_2^2]} = \theta_2 + \theta_1 a_{12} \frac{\sigma_1^2}{\sigma_2^2 + a_{12}^2 \sigma_1^2}.
\tag{20}
$$

Similarly, if we fit $X_2$ on $X_1$ then

$$
X_1 = \hat{a}_{12}^{-1} X_2 + \eta_2,
$$

then $\hat{a}_{12}^{-1} = \frac{\mathbb{E}[X_1 X_2]}{\mathbb{E}[X_2^2]}$. However $\mathbb{E}[X_1 X_2]$ can also be written as following:

$$
\mathbb{E}[X_1 X_2] = a_{12}^{-1}\mathbb{E}[X_2^2] - a_{12}^{-1}\mathbb{E}[\varepsilon_2 X_2].
$$

Hence,

$$
\hat{a}_{12}^{-1} = a_{12}^{-1}\left(1 - \frac{\sigma_2^2}{\sigma_2^2 + a_{12}^2 \sigma_1^2}\right) = a_{12}^{-1}\left(\frac{a_{12}^2 \sigma_1^2}{\sigma_2^2 + a_{12}^2 \sigma_1^2}\right).
$$

Residual $\hat{V} = X_1 - \hat{a}_{12}^{-1} X_2$. Hence we can have

$$
\mathbb{E}(\hat{V} X_1) = \mathbb{E}[X_1^2] - \hat{a}_{12}^{-1}\mathbb{E}[X_1 X_2] = \mathbb{E}[\varepsilon_1^2] - \hat{a}_{12}^{-1} a_{12}\mathbb{E}[\varepsilon_1^2] = \frac{\sigma_1^2 \sigma_2^2}{\sigma_2^2 + a_{12}^2 \sigma_1^2}.
$$

We now calculate,

$$\mathbb{E}\left[\hat{V}(Y - \hat{\theta}_2 X_2)\right] = \mathbb{E}\left[(X_1 - \hat{a}_{12}^{-1} X_2)(Y - \hat{\theta}_2 X_2)\right]$$

$$= \mathbb{E}\left[(X_1 - \hat{a}_{12}^{-1} X_2)\big((\theta_2 - \hat{\theta}_2)X_2 + \theta_1 X_1 + \varepsilon_3\big)\right]$$

$$= (\theta_2 - \hat{\theta}_2)a_{12}\sigma_1^2 + \theta_1\sigma_1^2 - \hat{a}_{12}^{-1}(\theta_2 - \hat{\theta}_2)(\sigma_2^2 + a_{12}^2\sigma_1^2) - \hat{a}_{12}^{-1}\theta_1 a_{12}\sigma_1^2$$

$$= \frac{\theta_1\sigma_1^2\sigma_2^2}{\sigma_2^2 + a_{12}^2\sigma_1^2}.$$

Last equation was written after a step of minor calculation. Since the estimator is

$$\hat{\theta}_1 = \left[\mathbb{E}(\hat{V}X_1)\right]^{-1}\mathbb{E}\left[\hat{V}(Y - \hat{\theta}_2 X_2)\right] = \theta_1.$$

$\square$

## A.2 Influence of the interactions between parents

In this section, we use a generic example shown in Figure 2 which we show again in Figure A.2 to illustrate the role of interactions between the covariates on the proposed causal discovery algorithm.

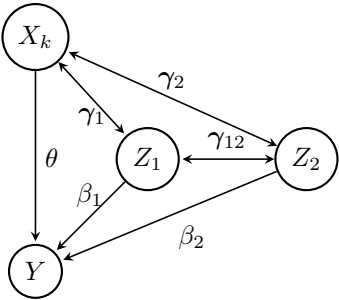

Figure A.2: A generic example of identification of a causal effect $\theta$ in the presence of causal and anti-causal interactions between the causal predictor and other putative parents, and possibly arbitrary cyclic and nonlinear assignments for all nodes except $Y$ (see Proposition 2). We have $X_{-k} = Z_1 \cup Z_2$.

The estimator discussed can simply be derived from the Neyman orthogonality condition. We now provide the below the proof for Proposition 2. For the sake of completeness, we also rewrite the statement of the proposition again.

**Proposition 6** (**Restatement of Proposition 2**). *Assume the structural causal model of Fig. A.2, with (possibly nonlinear and confounded) assignments between elements of $X = [X_k, X_{-k}^\top]^\top$, with $X_{-k} = [Z_1^\top, Z_2^\top]^\top$, parameterized by $\boldsymbol{\gamma} = (\gamma_1, \gamma_2, \gamma_{12})$. Assume the unconfounded linear structural assignment $Y := X_k\theta + X_{-k}^\top\boldsymbol{\beta} + U$, with $U$ zero mean random variable with finite variance $\sigma_U^2 > 0$, independent of $X$. Then, the score*

$$\psi(W; \theta, \boldsymbol{\beta}) = (Y - X_k\theta - X_{-k}^\top\boldsymbol{\beta})(X_k - r_{XX_{-k}}X_{-k}), \tag{21}$$

*with $r_{XX_{-k}} = \mathbb{E}[X_k X_{-k}^\top]\mathbb{E}[X_{-k}X_{-k}^\top]^{-1}$, follows the Neyman orthogonality condition for the estimation of $\theta$ with nuisance parameters $\boldsymbol{\eta} = (\beta, \boldsymbol{\gamma})$ which reads*

$$\mathbb{E}\left[(Y - X_k\theta - X_{-k}^\top\boldsymbol{\beta})(X_k - r_{XX_{-k}}X_{-k})\right] = 0.$$

*Proof of Proposition 2.* We first use a likelihood based approach under an additional Gaussianity assumption before addressing the general setting.

**Special case** Let us assume $U$ is Gaussian. Using the global Markov factorization for simple SCMs[8] (Forré & Mooij, 2017; Bongers et al., 2021),

$$P(W; \theta, \boldsymbol{\eta}) = P(Y | X_{-k}, X_k; \theta, \boldsymbol{\beta}) P(X_{-k}, X_k; \boldsymbol{\gamma}),$$

due to linearity and gaussianity of the assignment of $Y$, we obtain a negative log likelihood of the form (up to additive constants)

$$\ell(W; \theta, \boldsymbol{\eta}) = \frac{1}{2\sigma_U^2}(Y - X_k\theta - X_{-k}{}^\top\boldsymbol{\beta})(Y - X_k\theta - X_{-k}{}^\top\boldsymbol{\beta}) + f(X_k, X_{-k}; \boldsymbol{\gamma}).$$

where $f$ stands for the negative log likelihood of the second factor and $\boldsymbol{\eta} = [\boldsymbol{\beta}^\top, \boldsymbol{\gamma}^\top]^\top$ is the nuisance parameter vector. Following the principle of the approach described in main text, we use Chernozhukov et al. (2018a) [Eq. (2.7)] to define the Neyman orthogonalized score, leading to :

$$\psi(W; \theta, \boldsymbol{\eta}) = \partial_\theta \ell(W; (\theta, \boldsymbol{\eta})) - \boldsymbol{\mu}\partial_\eta \ell(W; (\theta, \boldsymbol{\eta})) = -\frac{1}{\sigma_U^2}(Y - X_k\theta - X_{-k}{}^\top\boldsymbol{\beta})X_k$$
$$- \boldsymbol{\mu} \begin{bmatrix} -\frac{1}{\sigma_U^2}(Y - X_k\theta - X_{-k}{}^\top\boldsymbol{\beta})X_{-k} \\ \partial_\gamma f(X_k, X_{-k}; \gamma) \end{bmatrix}.$$

The quantity $\boldsymbol{\mu}$ should be chosen to satisfy Neyman orthogonality of Equation (5), which leads to[9]

$$\partial_{\eta^\top} \mathbb{E}\psi(W; \theta, \boldsymbol{\eta}) = \partial_{\eta^\top}\mathbb{E}\partial_\theta \ell(W; (\theta, \boldsymbol{\eta})) - \boldsymbol{\mu}\partial_{\eta^\top}\mathbb{E}\partial_\eta \ell(W; (\theta, \boldsymbol{\eta})) = 0$$

leading to the expression of $\boldsymbol{\mu}$ given in Chernozhukov et al. (2018a) [eq. (2.8)]:

$$\boldsymbol{\mu} = J_{\theta,\eta}J_{\eta,\eta}^{-1},$$

with

$$J_{\eta,\eta} = \partial_{\eta^\top}\mathbb{E}\left[\partial_\eta \ell(W, \theta, \boldsymbol{\eta})\right] = \begin{bmatrix} \sigma_Y^{-2}\mathbb{E}\left[X_{-k}X_{-k}^\top\right] & \mathbf{0} \\ \mathbf{0} & \partial_{\gamma^\top}\mathbb{E}\left[\partial_\gamma f(X_k, X_{-k}; \boldsymbol{\gamma})\right] \end{bmatrix},$$

and

$$J_{\theta,\eta} = \partial_{\eta^\top}\mathbb{E}\left[\partial_\theta \ell(W, \theta, \boldsymbol{\eta})\right] = \sigma_U^{-2}\begin{bmatrix} \mathbb{E}\left[X_kX_{-k}^\top\right] & \mathbf{0} \end{bmatrix},$$

resulting in

$$\boldsymbol{\mu} = \begin{bmatrix} \mathbb{E}\left[X_kX_{-k}^\top\right]\mathbb{E}\left[X_{-k}X_{-k}^\top\right]^{-1}; \mathbf{0} \end{bmatrix} = \begin{bmatrix} r_{XX_{-k}}; \mathbf{0} \end{bmatrix}.$$

Reintroducing $\boldsymbol{\mu}$ in the expression of the correction term leads to

$$\boldsymbol{\mu}\partial_\eta \ell(W; (\theta, \boldsymbol{\eta})) = \boldsymbol{\mu}\begin{bmatrix} -\frac{1}{\sigma_U^2}(Y - X_k\theta - X_{-k}{}^\top\boldsymbol{\beta})X_{-k} \\ \partial_\gamma f(X_k, X_{-k}; \gamma) \end{bmatrix}.$$

leads to

$$\boldsymbol{\mu}\partial_\eta \ell(W; (\theta, \boldsymbol{\eta})) = -\frac{1}{\sigma_U^2}(Y - X_k\theta - X_{-k}{}^\top\boldsymbol{\beta})r_{XX_{-k}}X_{-k},$$

which leads to the final expression of the orthogonalized score $\psi$ :

$$\partial_\theta \ell(W; (\theta, \boldsymbol{\eta})) - \boldsymbol{\mu}\partial_\eta \ell(W; (\theta, \boldsymbol{\eta})) = -\frac{1}{\sigma_U^2}(Y - X_k\theta - X_{-k}{}^\top\boldsymbol{\beta})(X_k - r_{XX_{-k}}X_{-k}).$$

Since $\sigma_U$ is a positive constant of the problem, we can remove it while still having an appropriate orthogonalized score for this estimation problem, leading to the final expression

$$\psi(W; \theta, \boldsymbol{\eta}) = (Y - X_k\theta - X_{-k}{}^\top\boldsymbol{\beta})(X_k - r_{XX_{-k}}X_{-k}).$$

---

[8]The necessary condition for this statement to be true is uniquely solvability which is equivalent to not having self-cycles in the causal structure.

[9]the transpose in $\partial_{\eta^\top}$ should be understood as organizing columnwise the partial derivatives with respect to each components of $\eta$

**General case** We now drop the Gaussianity assumption. Interestingly, the least square objective

$$L(W; \theta, \boldsymbol{\eta}) = (Y - X_k\theta - {X_{-k}}^\top \boldsymbol{\beta})(Y - X_k\theta - {X_{-k}}^\top \boldsymbol{\beta}).$$

is still an appropriate (biased) score for the estimation problem, in which the nuisance parameters $\boldsymbol{\gamma}$ do not play a role. Indeed, starting from the structural equation of $Y$

$$Y \coloneqq X_k\theta + X_{-k}^\top \boldsymbol{\beta} + U,$$

we get

$$\mathbb{E}[YX_k] = \mathbb{E}[X_k^2]\theta + \mathbb{E}[X_{-k}^\top X_k]\boldsymbol{\beta},$$

which can be rewritten as

$$\mathbb{E}\partial_\theta L(W; (\theta_0, \boldsymbol{\eta}_0)) = 0.$$

As a consequence, the true parameter is the unique minimizer of the loss, identified by this vanishing paratial derivative. We can thus apply the previous orthogonalization procedure in the same way as we did for the likelihood, defining

$$\psi(W; \theta, \boldsymbol{\eta}) = \partial_\theta L(W; (\theta, \boldsymbol{\eta})) - \boldsymbol{\mu}\partial_\eta L(W; (\theta, \boldsymbol{\eta}))$$

and leading to the exact same orthogonalized score (up to irrelevant sign change).

$\square$

## A.3 Proof of proposition 3

*Proof.* From Equation (13)

$$\begin{aligned}
\mathbb{E}(Y \mid X_{-j}) &= \mathbb{E}(\langle\theta, X\rangle \mid X_{-j}) + \mathbb{E}(U \mid X_{-j}) = \mathbb{E}(\langle\theta_{-j}, X_{-j}\rangle \mid X_{-j}) + \theta_j\mathbb{E}(X_j \mid X_{-j}) \\
&= \langle\theta_{-j}, X_{-j}\rangle + \theta_j\mathbb{E}(X_j \mid X_{-j}) = \langle\theta, X\rangle - \theta_jX_j + \theta_j\mathbb{E}(X_j \mid X_{-j}) \\
&= Y - U - \theta_j(X_j - \mathbb{E}(X_j \mid X_{-j})).
\end{aligned}$$

Thus

$$\begin{aligned}
\chi_j &= \mathbb{E}\left[(U + \theta_j(X_j - \mathbb{E}(X_j \mid X_{-j}))) (X_j - \mathbb{E}(X_j \mid X_{-j})\right] \\
&= \mathbb{E}\left[U(X_j - \mathbb{E}(X_j \mid X_{-j})\right] + \theta_j\mathbb{E}\left[(X_j - \mathbb{E}(X_j \mid X_{-j})^2\right] \\
&= \theta_j\mathbb{E}\left[(X_j - \mathbb{E}(X_j \mid X_{-j})^2\right].
\end{aligned}$$

Since $\mathbb{E}\left[(X_j - \mathbb{E}(X_j \mid X_{-j})^2\right] > 0, j \in PA_Y$ if and only if $\chi_k \neq 0$, proving a)-b). For c), we rely on the properties of the Hilbert space of square integrable RV's $L^2(\Omega)$, equiped with the scalare product $\langle X, Y\rangle = \mathbb{E}[XY]$. We rewrite

$$\begin{aligned}
\chi_j &= \mathbb{E}\left[(Y - \mathbb{E}(Y \mid X_{-j})) \left(X_j - r_{XX_{-k}}X_{-j}\right)\right] \\
&+ \mathbb{E}\left[(Y - \mathbb{E}(Y \mid X_{-j})) \left(r_{XX_{-k}}X_{-j} - \mathbb{E}(X_j \mid X_{-j})\right)\right].
\end{aligned}$$

Under our assumptions, $\mathbb{E}(Y|X_{-j})$ is the orthogonal projection of $Y$ on the subspace of $\mathcal{G}$-measurable square integrable RV's $L^2(\Omega, \mathcal{G})$, so $Y - \mathbb{E}(Y|X_{-j})$ is orthogonal to any elements of $L^2(\Omega, \mathcal{G})$. Noticing that $\left(r_{XX_{-k}}X_{-j} - \mathbb{E}(X_j \mid X_{-j})\right)$ is an element of $L^2(\Omega, \mathcal{G})$, the second right-hand side term of the above equation vanishes and we get the result. $\square$

## A.4 Background on the Bilinear Influence Function (BIF) class and proof of Proposition 5

We summarize the results in Smucler et al. (2019) that justify the normal convergence of the Lasso-type estimates that we relying on in Section 2.5 to perform statistical hypothesis tests. This paper investigates the properties of $\ell_1$-regularised machine learning estimators for a particular family of non-parametric estimands, called Bilinear Influence Function (BIF) functionals.

The estimand is denoted $\chi(\eta)$ where $\eta$ denotes the model. The bias of the estimator $\hat{\eta}$ due to the unperfect estimate $\hat{\eta}$ of $\eta$ can be quantified using the influence function framework, leading to the following Taylor expansion in the neighborhood of the true model $\eta$,

$$\hat{\chi} = \chi(\hat{\eta}) + \mathbb{P}_n \chi_{\hat{\eta}}^1 ,$$

where $\chi_{\hat{\eta}}^1$ is the influence function, which is a mean zero random variable under distribution $P_\eta$ of the true model, and $\mathbb{P}_n$ is the empirical distribution of $n$ iid samples of the observation distribution.

Estimands belonging to the BIF class are characterized by an influence function of the following form, for observed random variables $O$ including a vector $Z$,

$$\chi_{\hat{\eta}}^1(O) = S_{ab} a(Z) b(Z) + m_a(O, a) + m_b(O, b) + S_0 - \chi(\eta)$$

where $a$ and $b$ are in $L_2$ and $m_a$ and $m_b$ are linear in the second argument (Smucler et al., 2019, Definition 1).

Our estimand of Equation (3) belongs to this BIF class as an expected conditional covariance (Smucler et al., 2019, Example 5), written in the notation of that paper as *Lin.L*, *Lin.E* and *Lin.V*.

$$\mathbb{E}[(Y - \mathbb{E}(Y|Z))(D - \mathbb{E}(D|Z))] .$$

For this case, Hines et al. (2022) (among others) provide the influence function:

$$\chi_{\hat{\eta}}^1(O) = (Y - \mathbb{E}(Y|Z))(D - \mathbb{E}(D|Z)) - \chi(\eta)$$

which entails the BIF function parameters $S_{ab} = 1$, $a(Z) = \mathbb{E}(Y|Z)$, $b(Z) = \mathbb{E}(D|Z)$, $m_a(O, a) = -Da$, $m_b(O, b) = -Yb$, $S_0 = DY$.

An estimation procedure of BIF functionals based on Lasso-type estimators of conditional expectations is described in Smucler et al. (2019, Section 3.1), and corresponds to ours described in section 2.5. In short, estimates are chosen among a family of models of the form $\varphi_a(\langle \theta_a, \phi(Z) \rangle)$ and $\varphi_b(\langle \theta_b, \phi(Z) \rangle)$, with parameters $\theta_c$, $c \in \{a, b\}$, that solve the $\ell_1$ regularized problem of the general form

$$\widehat{\theta}_c = \arg \min_{\theta \in \mathbb{R}^p} \mathbb{P}_n \left[ Q_c(\theta, \phi, w) \right] + \lambda \|\theta\|_1 , \tag{22}$$

for $c \in \{a, b\}$ with $\mathbb{P}_n$ an $n$-sample empirical average, $\lambda$ the regularization parameter and objective function

$$Qc(\theta; \phi; w) = S_{ab} w(Z) \psi_c(\langle \theta, \phi(Z) \rangle) + \langle \theta, m_{\bar{c}}(w \cdot \phi) \rangle$$

In our case, we can use linear models which entail an identity link function $\varphi = id$ and anti-derivative $\psi = x^2/1$. Moreover the weight can be set to $w = 1$ Smucler et al. (2019), leading to

$$Qa(\theta; \phi; w) = (\langle \theta, \phi(Z) \rangle)^2 - \langle \theta, Y\phi(Z) \rangle = (Y - \langle \theta, \phi(Z) \rangle)^2 - Y^2 ,$$

and

$$Q_b(\theta; \phi; w) = (\langle \theta, \phi(Z) \rangle)^2 - \langle \theta, D\phi(Z) \rangle = (D - \langle \theta, \phi(Z) \rangle)^2 - D^2 .$$

which thus both reduce to least square linear regression objectives, and thus turn eq. 22 into the classical Lasso objective.

Asymptotic properties are established for the true $a(Z)$ and/or $b(Z)$ belonging sequences of models associated to each sample size $n$ with parameter dimension $p$ and sparsity $s$, quantified as an upper bound on the number of non-zero coefficients of a vector, i.e. its $\ell_0$ norm $\|.\|_0$, such that

$$s \log(p)/n \underset{n \to +\infty}{\to} 0.$$

In particular, (Smucler et al., 2019, Section 4) use the *approximately generalized linear-sparse class*, for some $j$, such that there exists $\theta^* \in \mathbb{R}^p$ and a function $r(Z)$ satisfying (dependence on $n$ is dropped to ease notation)

$$c(Z) = \varphi(\langle \theta^*, \phi(Z) \rangle) + r(Z)$$

where $\|\theta^*\|_0 \leq s$ and $\mathbb{E}[r(Z)^2] \leq K(s \log(p)/n)^j$. In our main text Definition 4, we specialize this class to the linear case ($\phi = id$) and $j = 1$.

*Sketch of the proof of Proposition 5.* Statement (3) of Theorem 1 in Smucler et al. (2019) provides necessary conditions the estimators' error to converges to a zero mean distribution with an accurate empirical estimate of the the variance. These conditions are gathered in 3 subsets of assumptions called *Lin.L*, *Lin.E* and *Lin.V* (see conditions 1-3 in (Smucler et al., 2019, Section 5.1.1) together with statements of sufficient conditions to satisfy them. Based on these statements, we go through justifications for each assumptions.

**Lin.L**   Our assumption (i) entails that this as *Lin.L.1* trivially satisfied for $\mathbb{E}[Y|X_{-j}]$ (see (Smucler et al., 2019, Section 4, Example 9)). Moreover, *Lin.L.1* is satisfied for $\mathbb{E}[X_j|X_{-j}]$ explicitly by our assumption (ii).

For *Lin.L.2*, we use the statement of Smucler et al. (2019) that it can be replaced by an assumption of "tails decays at least as fast as the tails of an exponential random variable". This is achieved by the combined effect of our assumptions (iii) and (iv).

For *Lin.L.3-4*, we use the statement of Smucler et al. (2019) that our assumptions (iii) and (v) are sufficient.

*Lin.L.5*, is trivial as in our case $S_{ab} = 1$.

**Lin.E**   For *Lin.E.1* we use the statement by Smucler et al. (2019) that our assumption (vi) is sufficient.

For *Lin.E.2*, a) results from the (strictly positive) lower bound on the variances in our assumption (vi), combined with (iii-iv) for the requested upper bounds.

**Lin.V**   For *Lin.V.1* is directly stated in our assumption (iii).

For *Lin.V.2* is trivial as in our case $S_{ab} = 1$.

For *Lin.V.3* is directly imposed by our assumption (iii), as stated by Smucler et al. (2019) for their Example 9.

$\square$

## B  Examples

The result discussed in Proposition 2 is not directly intuitive. In simple words, there are two takeaways from Proposition 2: (i) the orthogonality condition remains invariant irrespective of the causal direction between $X_k$ and $Z$, and (ii) the second term in Proposition 6 suggests to use a linear estimator for modeling all the relations, given that the relation between $Z$ and $Y$ is linear.

To generate more intuition, we provide a few examples. Let us go back again to the three variable interaction assuming the following structural equation model:

$$
\begin{aligned}
Y &:= \theta_1 X_1 + \theta_2 X_2 + \varepsilon_3 \\
X_2 &:= f(X_1) + \varepsilon_2 \\
X_1 &:= \varepsilon_1,
\end{aligned}
\tag{23}
$$

where $f$ is a nonlinear function and $\varepsilon_1, \varepsilon_2$ and $\varepsilon_3$ are zero mean Gaussian noises.

- Consider the case when $f(x) = x^2$. The goal is to estimate the parameter $\theta_1$ which we call $\hat{\theta}_1$. We follow the standard double ML procedure assuming policy variable $X_1$ and control $X_2$, although the ground truth causal dependency $X_1 \to X_2$ in contradiction with such setting (see Equation (4)). The estimate of $\theta_2$ following the double ML procedure, which we call $\hat{\theta}_2 = \frac{\mathbb{E}[X_2 Y]}{\mathbb{E}[X_2^2]} = \theta_2 + \theta_1 \frac{\mathbb{E}[X_1 X_2]}{\mathbb{E}[X_2^2]}$. Similarly, we want to estimate $X_1 = \alpha X_2 + \eta$ from which we get, $\alpha = \frac{\mathbb{E}[X_1 X_2]}{\mathbb{E}[X_2]^2}$. It is easy to see that $\mathbb{E}[X_1 X_2] = \mathbb{E}[X_1^3] = 0$. Hence, $\alpha = 0$ and it is easy to see $\hat{\theta}_1 = \theta_1$.

- Consider now the more general case where $f$ is any nonlinear function. As in the previously discussed example, the goal is to estimate $\theta_1$. We have $\hat{\theta}_2 = \frac{\mathbb{E}[X_2 Y]}{\mathbb{E}[X_2^2]} = \theta_2 + \theta_1 \frac{\mathbb{E}[X_1 X_2]}{\mathbb{E}[X_2^2]}$. Similarly, $\alpha = \frac{\mathbb{E}[X_1 X_2]}{E[X_2^2]}$. We

substitute these estimates into the orthogonality condition in Proposition 6:

$$\mathbb{E}\left[(Y - X_1\hat{\theta}_1 - X_2\hat{\theta}_2)(X_1 - \alpha X_2)\right] = 0.$$

$$\Rightarrow \mathbb{E}\left[\left(Y - X_1\hat{\theta}_1 - X_2\hat{\theta}_2\right)\left(X_1 - \frac{\mathbb{E}[X_1 X_2]}{E[X_2^2]}X_2\right)\right] = 0.$$

$$\Rightarrow \mathbb{E}\left[\left(X_1(\theta_1 - \hat{\theta}_1) + (a_2 - \hat{\theta}_2)X_2 + \varepsilon_3\right)\right.$$
$$\left.\left(X_1 - \frac{\mathbb{E}[X_1 X_2]}{E[X_2^2]}X_2\right)\right] = 0.$$

$$\Rightarrow \hat{\theta}_1 = \theta_1.$$

From the above two examples, it is clear that even though the internal relations between the variables are nonlinear, all we need is an unbiased linear estimate to estimate the causal parameter.

## C  Data Generation and Evaluation Metric

### C.1  Data Generation

### C.1.1  Causal Structure Learning Data

For every combination of number of nodes (#nodes), connectivity $(p_s)$, noise level $(\sigma^2)$, number of observation $(n)$, and non-linear probability $(p_n)$ (look at Table C.1), 100 examples (DAGs) are generated and stored as csv files (altogether 72.000 DAGs are simulated, comprising a dataset of overall $>$10GB). For each DAG, $z$ number of samples are generated by sampling noise ($\epsilon$ in Equation (25)) with variance $\sigma^2$ starting from root of the DAG. For future benchmarking, the generated files will be made available with the code later on.

We generate DAGs (Direct Acyclic Graphs) in multiple steps: i) a random permutation of nodes is chosen as a topological order of a DAG. ii) Based on this order, directed edges are added to this DAG from each node to its followers with a certain probability $p_s$ (connectivity). iii) For each observation, values are assigned to nodes according to the topological order of the DAG in such a way that each node's value is determined by summing over transformations (linear or nonlinear with a certain nonlinear probability $p_n$) of values of its direct causes with the addition of Gaussian distributed noise. The non-linear transformation used is $a\tanh(bx)$[10], with $a = 0.5$ and $b = 1.5$. If the set of parents for the node $X'$ is denoted as $PA_{X'}$ as before then value assignment for a node $X'$ is as follow:

$$X' = \varepsilon + \sum_{X \in PA_{X'}} \iota_\ell(p_n)\theta X + (1 - \iota_\ell(p_n)).a.\tanh(bX), \tag{24}$$

where $\varepsilon \sim N(0, \sigma^2)$ in which $\sigma^2$ represents noise level. $\iota_\ell(X)$ is an indicator functions which decides between linear or non-linear contribution of $X$ in $X'$. We decide the value of $\iota_\ell(p_n)$ by generating a binary randon number which is

---

[10]The resulting values in the experiments are not concentrated around zero, and they are even up to 10ks for large graphs ($\sim$ 50 nodes). With the nonlinearity feature of $a\tanh(bx)$ for relatively large values taken into account, this is a good representer of nonlinear relationships.

| connectivity $(p_s)$ | # nodes | nonlinear probability $(p_n)$ | # observ. n | noise level $(\sigma^2)$ |
|---|---|---|---|---|
| 0.1 | 5 | 0 | 100 | 0.01 |
| 0.3 | 10 | 0.3 | 500 | 0.1 |
| 0.5 | 20 | 0.5 | 1.000 | 0.3 |
| | 50 | 1 | | 0.5 |
| | | | | 1 |

Table C.1: Experimental Setup: In the experiments we vary the connectivity parameter, the number of nodes in the graph, the non-linear probability, the number of observations and the noise level and generate 100 graphs for each setting.

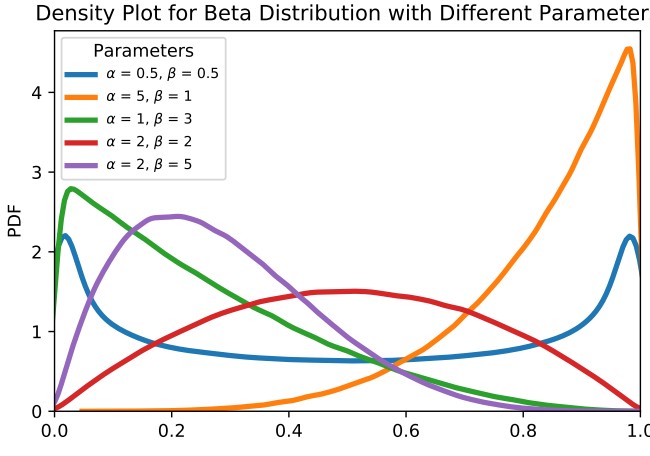

Figure C.1: Beta distribution with different parameters.

1 with probablity $p_n$ and 0 with probability $1 - p_n$. The value of $\theta$ is set to 2 for the small DAGs (number of nodes equal to 5 or 10) and 0.5 for large DAGs (number of nodes equal to 20 or 50) due to the value exploitation that might happen in large graphs.

We vary and investigate the effect of non-linear relationships, the number of nodes, number of observations, effect of connectivity and noise level while simulating the data. We summarize the factors in the data generation in Table C.1.

### C.1.2 Inference by Regression Data

Similar to Appendix C.1.1, data generation follows the random topological order but with a slight difference, i.e., with the following value assignment,

$$X' = \varepsilon + \sum_{X \in PA_{X'}} \iota_\ell(p_n)\theta X + (1 - \iota_\ell(p_n)).a.\tanh(bX), \tag{25}$$

where $\varepsilon \sim \mathcal{B}(\alpha, \beta)$ in which $\alpha$ and $\beta$ are parameters of beta distribution. The reason for this is that most of the inference methods exploit normality tests and this way it is possible to challenge them. A diverse set of parameters are chosen for the beta distribution to signifies this point (see Figure C.1). The set of parameters of DAGs varies according to Table C.2. For each setting, 50 examples are generated and stored (15000 DAGs overall) which will be available for future studies.

| connectivity $(p_s)$ | # nodes | nonlinear probability $(p_n)$ | # observ. n | beta distribution parameter $(\alpha, \beta)$ |
|---|---|---|---|---|
| 0.1 | 100 | 0 | 10 | (0.5, 0.5) |
| 0.3 | | 0.3 | 20 | (5, 1) |
| 0.5 | | 0.5 | 50 | (1, 3) |
| | | 1 | 100 | (2, 2) |
| | | | 200 | (2, 5) |

Table C.2: Experimental Setup: In the experiments we vary the connectivity parameter, the non-linear probability, the number of observations and beta distribution parameters. We generate 50 graphs for each setting.

**Code:** The code for the method and data generation is available in this GitHub repository.

## C.2   Evaluation Metric

Correctly and incorrectly inferred direct causes are considered true and false. Let the total number of true positives, false positives, true negatives ,and false negatives denoted by TP, FP, TN, and FN, we evaluate our method using following metrics:

- Recall (true positive rate):

$$TPR = \frac{TP}{TP + FN}$$

- Fall-out (false positive rate):

$$FPR = \frac{FP}{FP + TN}$$

- Critical Success Index (CSI): also known as Threat Score.

$$CSI = \frac{TP}{TP + FN + FP}$$

- Accuracy:

$$ACC = \frac{TP + TN}{P + N}$$

- F1 Score: harmonic mean of precision and sensitivity.

$$F_1 = \frac{2TP}{2TP + FP + FN}$$

- Matthews correlation coefficient (MCC): a metric for evaluating quality of binary classification introduced in (Matthews, 1975).

$$MCC = \frac{TP \times TN - FP \times FN}{\sqrt{(TP + FP)(TP + FN)(TN + FP)(TN + FN)}}$$

In some rare cases, we encountered zero-divided-by-zero and divided-by-zero cases for some of these metrics. In these situations, scores are reported 1 and 0 respectively while Fall-out is reported 0 and 1.

## C.3   Supplementary Tables for Performance in Inferring Direct Causes

Here, additional tables for the result of the experiments are provided.

### C.3.1   Compared to Causal Structure Learning Methods

In this section, supplementary tables supporting superior performance of CORTH Features compared to well-established Causal and Markov Blanket discovery methods are provided (See Tables C.3 to C.7). This superiority is consistent w.r.t. connectivity (Table C.5), number of nodes (Table C.3), number of observations (Table C.7), nonlinearity (Table C.4), and noise (Table C.6) using different evaluation metrics.

### C.3.2   Compared to Inference for Regression Methods

In this part, the superiority of our method in comparison to decent Inference for Regression methods, in different settings of connectivity (Table C.10), beta distribution parameters (Table C.11), number of observations (Table C.9), and nonlinearity (Table C.8) is provided.

Table C.3: Performance across all the settings for different number of nodes. Each single entry in the table is averaged over 18000 simulations. Our method is almost state of the art in every case.

| | Number of Nodes | | | | | | | | | | | |
| | 5 | | | 10 | | | 20 | | | 50 | | |
| Method | ACC | CSI | F1 | ACC | CSI | F1 | ACC | CSI | F1 | ACC | CSI | F1 |
|---|---|---|---|---|---|---|---|---|---|---|---|---|
| GES | 0.935 | 0.890 | 0.911 | 0.854 | 0.730 | 0.779 | 0.743 | 0.442 | 0.526 | 0.698 | 0.245 | 0.323 |
| rankGES | 0.923 | 0.857 | 0.883 | 0.846 | 0.700 | 0.753 | 0.740 | 0.428 | 0.514 | 0.697 | 0.237 | 0.316 |
| ARGES | 0.922 | 0.864 | 0.885 | 0.797 | 0.551 | 0.584 | 0.752 | 0.447 | 0.524 | 0.705 | 0.186 | 0.221 |
| rankARGES | 0.914 | 0.838 | 0.861 | 0.793 | 0.537 | 0.572 | 0.750 | 0.435 | 0.514 | 0.705 | 0.181 | 0.216 |
| FCI+ | 0.963 | 0.918 | 0.932 | 0.873 | 0.744 | 0.808 | 0.830 | 0.602 | 0.703 | 0.766 | 0.368 | 0.486 |
| LINGAM | **0.991** | **0.978** | **0.982** | **0.953** | 0.865 | 0.889 | 0.891 | 0.712 | 0.778 | 0.750 | 0.318 | 0.385 |
| PC | 0.957 | 0.913 | 0.929 | 0.864 | 0.723 | 0.786 | 0.823 | 0.569 | 0.664 | 0.763 | 0.348 | 0.457 |
| rankPC | 0.946 | 0.891 | 0.912 | 0.854 | 0.701 | 0.768 | 0.813 | 0.541 | 0.638 | 0.754 | 0.324 | 0.431 |
| MMPC | 0.868 | 0.586 | 0.597 | 0.823 | 0.494 | 0.535 | 0.790 | 0.412 | 0.489 | 0.749 | 0.260 | 0.350 |
| MMHC | 0.929 | 0.878 | 0.905 | 0.841 | 0.675 | 0.739 | 0.767 | 0.432 | 0.507 | 0.725 | 0.218 | 0.281 |
| GS | 0.883 | 0.613 | 0.623 | 0.855 | 0.563 | 0.601 | 0.824 | 0.501 | 0.580 | 0.759 | 0.310 | 0.388 |
| IAMB | 0.850 | 0.571 | 0.585 | 0.791 | 0.508 | 0.561 | 0.806 | 0.500 | 0.587 | 0.768 | 0.337 | 0.424 |
| FastIAMB | 0.883 | 0.614 | 0.624 | 0.858 | 0.571 | 0.611 | 0.828 | 0.511 | 0.593 | 0.770 | 0.326 | 0.409 |
| IAMB-FDR | 0.869 | 0.584 | 0.593 | 0.831 | 0.494 | 0.526 | 0.825 | 0.484 | 0.558 | 0.770 | 0.322 | 0.406 |
| PCI | 0.984 | 0.965 | 0.972 | 0.922 | 0.844 | 0.875 | 0.888 | 0.734 | 0.782 | 0.773 | 0.414 | 0.491 |
| Lasso | 0.965 | 0.948 | 0.968 | 0.905 | 0.834 | 0.892 | 0.894 | 0.786 | 0.866 | 0.773 | 0.489 | 0.627 |
| CORTH Features (Ours) | 0.988 | 0.968 | 0.973 | 0.949 | **0.908** | **0.934** | **0.949** | **0.865** | **0.905** | **0.795** | **0.559** | **0.663** |

| | Number of Nodes | | | | | | | | | | | |
| | 5 | | | 10 | | | 20 | | | 50 | | |
| Method | TPR | FPR | MCC | TPR | FPR | MCC | TPR | FPR | MCC | TPR | FPR | MCC |
|---|---|---|---|---|---|---|---|---|---|---|---|---|
| GES | 0.934 | 0.056 | 0.891 | 0.790 | 0.090 | 0.711 | 0.502 | 0.088 | 0.436 | 0.304 | 0.083 | 0.221 |
| rankGES | 0.924 | 0.068 | 0.877 | 0.780 | 0.098 | 0.695 | 0.493 | 0.089 | 0.425 | 0.297 | 0.083 | 0.215 |
| ARGES | 0.903 | 0.046 | 0.906 | 0.590 | 0.041 | 0.841 | 0.500 | 0.073 | 0.557 | 0.220 | 0.020 | 0.794 |
| rankARGES | 0.897 | 0.054 | 0.896 | 0.584 | 0.044 | 0.832 | 0.495 | 0.075 | 0.549 | 0.216 | 0.020 | 0.789 |
| FCI+ | 0.969 | 0.029 | 0.948 | 0.797 | 0.054 | 0.759 | 0.642 | 0.042 | 0.645 | 0.389 | 0.030 | 0.454 |
| LINGAM | 0.991 | 0.007 | 0.988 | 0.886 | 0.008 | 0.934 | 0.770 | 0.055 | 0.759 | 0.391 | 0.072 | 0.471 |
| PC | 0.950 | 0.024 | 0.941 | 0.759 | 0.041 | 0.759 | 0.600 | 0.032 | 0.650 | 0.363 | 0.021 | 0.468 |
| rankPC | 0.944 | 0.039 | 0.925 | 0.750 | 0.053 | 0.734 | 0.580 | 0.034 | 0.629 | 0.341 | 0.024 | 0.427 |
| MMPC | 0.588 | 0.006 | 0.965 | 0.498 | 0.011 | 0.852 | 0.417 | 0.006 | 0.684 | 0.261 | 0.003 | 0.528 |
| MMHC | 0.895 | 0.011 | 0.903 | 0.691 | 0.015 | 0.724 | 0.444 | 0.009 | 0.523 | 0.219 | 0.005 | 0.330 |
| GS | 0.615 | 0.002 | 0.973 | 0.566 | 0.001 | 0.895 | 0.506 | 0.001 | 0.739 | 0.311 | 0.000 | 0.688 |
| IAMB | 0.573 | 0.003 | 0.960 | 0.511 | 0.002 | 0.848 | 0.505 | 0.001 | 0.711 | 0.338 | 0.001 | 0.660 |
| FastIAMB | 0.616 | 0.003 | 0.972 | 0.575 | 0.002 | 0.888 | 0.518 | 0.002 | 0.734 | 0.327 | 0.001 | 0.677 |
| IAMB-FDR | 0.585 | 0.001 | 0.975 | 0.494 | 0.001 | 0.909 | 0.485 | 0.001 | 0.766 | 0.322 | 0.001 | 0.661 |
| PCI | 0.992 | 0.017 | 0.981 | 0.875 | 0.028 | 0.890 | 0.754 | 0.016 | 0.839 | 0.430 | 0.030 | 0.638 |
| Lasso | 0.999 | 0.074 | 0.949 | 0.944 | 0.119 | 0.817 | 0.954 | 0.147 | 0.794 | 0.681 | 0.148 | 0.488 |
| CORTH Features (Ours) | 0.999 | 0.016 | 0.986 | 0.952 | 0.044 | 0.906 | 0.884 | 0.011 | 0.894 | 0.609 | 0.101 | 0.567 |

Table C.4: Performance across all the settings for different number of nonlinear probabilities. Each single entry in the table is averaged over 18000 simulations. Our method is almost state of the art in every case.

| | Nonlinear Probability | | | | | | | | | | | |
|---|---|---|---|---|---|---|---|---|---|---|---|---|
| Method | 0 | | | 0.3 | | | 0.5 | | | 1 | | |
| | ACC | CSI | F1 | ACC | CSI | F1 | ACC | CSI | F1 | ACC | CSI | F1 |
| GES | 0.803 | 0.583 | 0.646 | 0.806 | 0.566 | 0.622 | 0.811 | 0.577 | 0.632 | 0.810 | 0.581 | 0.641 |
| rankGES | 0.796 | 0.559 | 0.625 | 0.801 | 0.546 | 0.605 | 0.805 | 0.556 | 0.613 | 0.805 | 0.561 | 0.623 |
| ARGES | 0.781 | 0.476 | 0.515 | 0.786 | 0.486 | 0.525 | 0.792 | 0.506 | 0.546 | 0.818 | 0.581 | 0.628 |
| rankARGES | 0.778 | 0.461 | 0.503 | 0.782 | 0.474 | 0.515 | 0.788 | 0.490 | 0.531 | 0.814 | 0.564 | 0.615 |
| FCI+ | 0.827 | 0.599 | 0.674 | 0.860 | 0.663 | 0.745 | 0.872 | 0.685 | 0.764 | 0.873 | 0.685 | 0.746 |
| LINGAM | **0.907** | 0.738 | 0.778 | 0.886 | 0.689 | 0.725 | 0.880 | 0.684 | 0.724 | 0.911 | 0.762 | 0.808 |
| PC | 0.818 | 0.574 | 0.641 | 0.854 | 0.641 | 0.720 | 0.864 | 0.665 | 0.7430 | 0.869 | 0.672 | 0.731 |
| rankPC | 0.813 | 0.560 | 0.630 | 0.841 | 0.614 | 0.694 | 0.848 | 0.627 | 0.704 | 0.864 | 0.656 | 0.720 |
| MMPC | 0.775 | 0.372 | 0.416 | 0.809 | 0.439 | 0.503 | 0.818 | 0.462 | 0.528 | 0.828 | 0.479 | 0.523 |
| MMHC | 0.797 | 0.516 | 0.578 | 0.815 | 0.549 | 0.610 | 0.823 | 0.566 | 0.625 | 0.826 | 0.571 | 0.620 |
| GS | 0.806 | 0.450 | 0.491 | 0.828 | 0.494 | 0.554 | 0.835 | 0.510 | 0.571 | 0.851 | 0.534 | 0.576 |
| IAMB | 0.762 | 0.389 | 0.440 | 0.799 | 0.463 | 0.538 | 0.809 | 0.488 | 0.565 | 0.830 | 0.520 | 0.576 |
| FastIAMB | 0.807 | 0.454 | 0.497 | 0.835 | 0.503 | 0.566 | 0.842 | 0.522 | 0.587 | 0.855 | 0.543 | 0.587 |
| IAMB-FDR | 0.796 | 0.418 | 0.457 | 0.818 | 0.456 | 0.511 | 0.827 | 0.481 | 0.538 | 0.853 | 0.529 | 0.578 |
| PCI | 0.853 | 0.674 | 0.720 | 0.897 | 0.746 | 0.789 | 0.905 | 0.763 | 0.806 | 0.911 | 0.774 | 0.805 |
| Lasso | 0.847 | 0.694 | 0.773 | 0.891 | 0.776 | 0.853 | 0.902 | 0.797 | 0.869 | 0.896 | 0.790 | 0.857 |
| CORTH Features (Ours) | 0.871 | **0.768** | **0.824** | **0.934** | **0.830** | **0.873** | **0.943** | **0.851** | **0.891** | **0.933** | **0.852** | **0.887** |
| | Nonlinear Probability | | | | | | | | | | | |
| Method | 0 | | | 0.3 | | | 0.5 | | | 1 | | |
| | TPR | FPR | MCC | TPR | FPR | MCC | TPR | FPR | MCC | TPR | FPR | MCC |
| GES | 0.643 | 0.093 | 0.564 | 0.620 | 0.074 | 0.557 | 0.629 | 0.071 | 0.568 | 0.637 | 0.079 | 0.570 |
| rankGES | 0.633 | 0.100 | 0.550 | 0.612 | 0.080 | 0.546 | 0.620 | 0.076 | 0.557 | 0.628 | 0.083 | 0.559 |
| ARGES | 0.514 | 0.041 | 0.789 | 0.526 | 0.041 | 0.793 | 0.547 | 0.043 | 0.791 | 0.626 | 0.055 | 0.725 |
| rankARGES | 0.509 | 0.044 | 0.780 | 0.522 | 0.044 | 0.788 | 0.540 | 0.046 | 0.783 | 0.620 | 0.059 | 0.715 |
| FCI+ | 0.638 | 0.045 | 0.637 | 0.704 | 0.037 | 0.708 | 0.728 | 0.035 | 0.731 | 0.728 | 0.037 | 0.730 |
| LINGAM | 0.775 | 0.025 | 0.832 | 0.723 | 0.028 | 0.759 | 0.722 | 0.034 | 0.741 | 0.819 | 0.053 | 0.822 |
| PC | 0.605 | 0.037 | 0.649 | 0.672 | 0.027 | 0.707 | 0.695 | 0.025 | 0.728 | 0.702 | 0.029 | 0.734 |
| rankPC | 0.597 | 0.043 | 0.626 | 0.656 | 0.040 | 0.680 | 0.668 | 0.036 | 0.695 | 0.692 | 0.031 | 0.714 |
| MMPC | 0.376 | 0.009 | 0.754 | 0.442 | 0.006 | 0.738 | 0.465 | 0.005 | 0.749 | 0.482 | 0.005 | 0.787 |
| MMHC | 0.528 | 0.017 | 0.581 | 0.561 | 0.008 | 0.623 | 0.578 | 0.007 | 0.636 | 0.582 | 0.008 | 0.639 |
| GS | 0.452 | 0.001 | 0.850 | 0.496 | 0.001 | 0.797 | 0.513 | 0.001 | 0.799 | 0.538 | 0.001 | 0.849 |
| IAMB | 0.847 | 0.003 | 0.773 | 0.891 | 0.002 | 0.853 | 0.902 | 0.001 | 0.869 | 0.896 | 0.001 | 0.857 |
| FastIAMB | 0.457 | 0.001 | 0.848 | 0.506 | 0.002 | 0.784 | 0.526 | 0.002 | 0.791 | 0.548 | 0.002 | 0.848 |
| IAMB-FDR | 0.418 | 0.001 | 0.837 | 0.457 | 0.001 | 0.819 | 0.481 | 0.001 | 0.822 | 0.530 | 0.001 | 0.833 |
| PCI | 0.712 | 0.057 | 0.792 | 0.765 | 0.011 | 0.848 | 0.781 | 0.010 | 0.845 | 0.794 | 0.013 | 0.864 |
| Lasso | 0.823 | 0.130 | 0.684 | 0.907 | 0.120 | 0.778 | 0.926 | 0.116 | 0.800 | 0.921 | 0.122 | 0.787 |
| CORTH Features (Ours) | 0.840 | 0.119 | 0.730 | 0.849 | 0.007 | 0.872 | 0.870 | 0.008 | 0.888 | 0.885 | 0.038 | 0.863 |

Table C.5: Performance across all the settings for different connectivities. Each single entry in the table is averaged over 24000 simulations. Our method is almost state of the art in every case.

| Method | Connectivity | | | | | | | | | | | |
| | 0.1 | | | | 0.3 | | | | 0.5 | | | |
| | ACC | CSI | F1 | MCC | ACC | CSI | F1 | MCC | ACC | CSI | F1 | MCC |
|---|---|---|---|---|---|---|---|---|---|---|---|---|
| GES | 0.961 | 0.786 | 0.825 | 0.857 | 0.815 | 0.539 | 0.598 | 0.522 | 0.646 | 0.405 | 0.482 | 0.315 |
| rankGES | 0.954 | 0.746 | 0.790 | 0.840 | 0.809 | 0.522 | 0.584 | 0.511 | 0.642 | 0.398 | 0.475 | 0.308 |
| ARGES | 0.965 | 0.794 | 0.828 | 0.876 | 0.805 | 0.456 | 0.501 | 0.726 | 0.612 | 0.286 | 0.330 | 0.720 |
| rankARGES | 0.959 | 0.763 | 0.801 | 0.863 | 0.802 | 0.447 | 0.494 | 0.721 | 0.611 | 0.282 | 0.328 | 0.716 |
| FCI+ | 0.974 | 0.819 | 0.853 | 0.910 | 0.866 | 0.631 | 0.714 | 0.674 | 0.734 | 0.524 | 0.629 | 0.521 |
| LINGAM | 0.966 | 0.763 | 0.796 | 0.889 | 0.896 | 0.710 | 0.753 | 0.761 | 0.827 | 0.682 | 0.727 | 0.715 |
| PC | 0.975 | 0.819 | 0.849 | 0.921 | 0.861 | 0.609 | 0.689 | 0.676 | 0.718 | 0.486 | 0.588 | 0.516 |
| rankPC | 0.971 | 0.797 | 0.831 | 0.912 | 0.852 | 0.587 | 0.670 | 0.653 | 0.701 | 0.458 | 0.560 | 0.470 |
| MMPC | 0.949 | 0.606 | 0.637 | 0.901 | 0.815 | 0.390 | 0.451 | 0.722 | 0.658 | 0.318 | 0.389 | 0.648 |
| MMHC | 0.978 | 0.834 | 0.867 | 0.901 | 0.830 | 0.497 | 0.561 | 0.574 | 0.639 | 0.321 | 0.397 | 0.385 |
| GS | 0.954 | 0.644 | 0.669 | 0.935 | 0.843 | 0.467 | 0.524 | 0.815 | 0.693 | 0.380 | 0.451 | 0.722 |
| IAMB | 0.969 | 0.692 | 0.745 | 0.864 | 0.841 | 0.463 | 0.522 | 0.807 | 0.692 | 0.377 | 0.452 | 0.709 |
| FastIAMB | 0.955 | 0.650 | 0.676 | 0.931 | 0.845 | 0.474 | 0.535 | 0.804 | 0.705 | 0.392 | 0.467 | 0.718 |
| IAMB-FDR | 0.950 | 0.608 | 0.626 | 0.961 | 0.832 | 0.436 | 0.492 | 0.816 | 0.689 | 0.369 | 0.446 | 0.707 |
| PCI | 0.986 | 0.902 | 0.920 | 0.954 | 0.906 | 0.716 | 0.759 | **0.838** | 0.783 | 0.600 | 0.661 | 0.720 |
| Lasso | 0.976 | 0.886 | 0.925 | 0.926 | 0.876 | 0.725 | 0.811 | 0.737 | 0.800 | 0.682 | 0.778 | 0.622 |
| CORTH Features (Ours) | **0.988** | **0.915** | **0.934** | **0.959** | **0.926** | **0.813** | **0.858** | 0.833 | **0.847** | **0.747** | **0.814** | **0.724** |

Table C.6: Performance across all the settings for different noise levels. Each single entry in the table is averaged over 14400 simulations. Our method is almost state of the art in every case.

| Method | Noise Level | | | | | | | | | | | |
| | 0.01 | | | | 0.5 | | | | 1 | | | |
| | ACC | CSI | F1 | MCC | ACC | CSI | F1 | MCC | ACC | CSI | F1 | MCC |
|---|---|---|---|---|---|---|---|---|---|---|---|---|
| GES | 0.804 | 0.579 | 0.639 | 0.559 | 0.808 | 0.571 | 0.629 | 0.562 | 0.818 | 0.586 | 0.644 | 0.589 |
| rankGES | 0.797 | 0.557 | 0.619 | 0.548 | 0.802 | 0.552 | 0.613 | 0.551 | 0.812 | 0.565 | 0.625 | 0.577 |
| ARGES | 0.810 | 0.572 | 0.625 | 0.653 | 0.789 | 0.496 | 0.534 | 0.814 | 0.774 | 0.434 | 0.460 | 0.897 |
| rankARGES | 0.804 | 0.549 | 0.605 | 0.643 | 0.786 | 0.483 | 0.523 | 0.806 | 0.774 | 0.433 | 0.459 | 0.895 |
| FCI+ | 0.843 | 0.617 | 0.691 | 0.674 | 0.865 | 0.678 | 0.753 | 0.717 | 0.874 | 0.697 | 0.766 | 0.740 |
| LINGAM | 0.888 | 0.703 | 0.744 | 0.763 | 0.899 | 0.723 | 0.763 | 0.797 | 0.903 | 0.732 | 0.773 | 0.803 |
| PC | 0.837 | 0.595 | 0.664 | 0.683 | 0.859 | 0.659 | 0.731 | 0.716 | 0.870 | 0.686 | 0.752 | 0.745 |
| rankPC | 0.831 | 0.584 | 0.657 | 0.653 | 0.845 | 0.626 | 0.699 | 0.688 | 0.856 | 0.655 | 0.724 | 0.714 |
| MMPC | 0.796 | 0.405 | 0.456 | 0.762 | 0.812 | 0.453 | 0.510 | 0.756 | 0.825 | 0.480 | 0.533 | 0.780 |
| MMHC | 0.806 | 0.526 | 0.585 | 0.605 | 0.818 | 0.557 | 0.615 | 0.626 | 0.829 | 0.586 | 0.639 | 0.652 |
| GS | 0.820 | 0.468 | 0.518 | **0.824** | 0.836 | 0.513 | 0.566 | 0.819 | 0.846 | 0.538 | 0.586 | 0.833 |
| IAMB | 0.784 | 0.421 | 0.483 | 0.779 | 0.807 | 0.485 | 0.552 | 0.769 | 0.823 | 0.523 | 0.586 | 0.790 |
| FasIAMB | 0.821 | 0.469 | 0.520 | 0.819 | 0.842 | 0.526 | 0.582 | 0.814 | 0.852 | 0.548 | 0.600 | 0.828 |
| IAMB-FDR | 0.810 | 0.432 | 0.478 | 0.834 | 0.831 | 0.492 | 0.545 | 0.825 | 0.841 | 0.514 | 0.563 | 0.841 |
| PCI | 0.873 | 0.690 | 0.730 | 0.819 | 0.901 | 0.760 | 0.801 | 0.846 | 0.906 | 0.777 | 0.815 | 0.854 |
| Lasso | 0.868 | 0.728 | 0.807 | 0.725 | 0.891 | 0.780 | 0.852 | 0.779 | 0.898 | 0.794 | 0.861 | 0.793 |
| CORTH Features (Ours) | **0.899** | **0.789** | **0.839** | 0.795 | **0.929** | **0.842** | **0.883** | **0.858** | **0.934** | **0.854** | **0.891** | **0.866** |

Table C.7: Performance across all the settings for different number of observations. Each single entry in the table is averaged over 24000 simulations. Our method is almost state of the art in every case.

| Method | Number of Observations | | | | | | | | | | | |
|---|---|---|---|---|---|---|---|---|---|---|---|---|
| | 100 | | | | 500 | | | | 1000 | | | |
| | ACC | CSI | F1 | MCC | ACC | CSI | F1 | MCC | ACC | CSI | F1 | MCC |
| GES | 0.797 | 0.524 | 0.588 | 0.539 | 0.811 | 0.593 | 0.650 | 0.572 | 0.815 | 0.612 | 0.666 | 0.583 |
| rankGES | 0.788 | 0.495 | 0.561 | 0.522 | 0.806 | 0.576 | 0.636 | 0.564 | 0.810 | 0.595 | 0.652 | 0.573 |
| ARGES | 0.780 | 0.446 | 0.489 | 0.786 | 0.799 | 0.535 | 0.576 | 0.773 | 0.803 | 0.555 | 0.595 | 0.764 |
| rankARGES | 0.776 | 0.428 | 0.473 | 0.778 | 0.795 | 0.523 | 0.566 | 0.766 | 0.800 | 0.542 | 0.584 | 0.757 |
| FCI+ | 0.837 | 0.589 | 0.671 | 0.652 | 0.865 | 0.684 | 0.755 | 0.720 | 0.871 | 0.702 | 0.771 | 0.732 |
| LINGAM | 0.840 | 0.578 | 0.650 | 0.678 | 0.908 | 0.719 | 0.743 | 0.825 | 0.941 | 0.858 | 0.883 | 0.862 |
| PC | 0.830 | 0.568 | 0.642 | 0.661 | 0.858 | 0.662 | 0.732 | 0.719 | 0.866 | 0.684 | 0.752 | 0.733 |
| rankPC | 0.821 | 0.544 | 0.617 | 0.632 | 0.849 | 0.639 | 0.711 | 0.696 | 0.855 | 0.660 | 0.733 | 0.707 |
| MMPC | 0.771 | 0.323 | 0.368 | 0.787 | 0.819 | 0.476 | 0.534 | 0.739 | 0.832 | 0.515 | 0.575 | 0.745 |
| MMHC | 0.800 | 0.495 | 0.557 | 0.579 | 0.820 | 0.570 | 0.625 | 0.633 | 0.826 | 0.587 | 0.642 | 0.647 |
| GS | 0.793 | 0.375 | 0.427 | 0.785 | 0.842 | 0.540 | 0.592 | 0.834 | 0.856 | 0.577 | 0.625 | 0.852 |
| IAMB | 0.745 | 0.316 | 0.390 | 0.705 | 0.815 | 0.510 | 0.574 | 0.794 | 0.835 | 0.556 | 0.614 | 0.821 |
| FastIAMB | 0.803 | 0.401 | 0.461 | 0.770 | 0.844 | 0.541 | 0.593 | 0.833 | 0.857 | 0.574 | 0.623 | 0.850 |
| IAMB-FDR | 0.783 | 0.325 | 0.372 | 0.825 | 0.837 | 0.523 | 0.578 | 0.815 | 0.850 | 0.564 | 0.613 | 0.843 |
| PCI | 0.829 | 0.551 | 0.594 | **0.804** | 0.914 | 0.812 | 0.853 | 0.842 | 0.931 | 0.855 | 0.893 | 0.866 |
| Lasso | 0.870 | **0.729** | **0.812** | 0.732 | 0.889 | 0.778 | 0.848 | 0.773 | 0.893 | 0.786 | 0.854 | 0.780 |
| CORTH Features (Ours) | **0.883** | 0.710 | 0.780 | 0.754 | **0.935** | **0.874** | **0.906** | **0.874** | **0.942** | **0.891** | **0.920** | **0.887** |

Table C.8: Performance across all the settings for different nonlinear probabilities. Each single entry in the table is averaged over 3750 simulations.

| Method | Nonlinear Probability | | | | | | | | | | | |
|---|---|---|---|---|---|---|---|---|---|---|---|---|
| | 0 | | | 0.3 | | | 0.5 | | | 1 | | |
| | TPR | CSI | F1 | TPR | CSI | F1 | TPR | CSI | F1 | TPR | CSI | F1 |
| Standard Regression | 0.149 | 0.103 | 0.139 | 0.166 | 0.112 | 0.141 | 0.175 | 0.108 | 0.136 | 1.000 | 0.801 | 0.801 |
| Lasso | 0.237 | 0.116 | 0.176 | 0.285 | 0.126 | 0.202 | 0.360 | 0.165 | 0.263 | 1.000 | 0.046 | 0.046 |
| Debiased Lasso | 0.238 | 0.117 | 0.178 | 0.267 | 0.112 | 0.178 | 0.300 | 0.124 | 0.202 | 1.000 | 0.050 | 0.050 |
| Forward Stepwise Reg_active | 0.174 | 0.112 | 0.162 | 0.157 | 0.110 | 0.167 | 0.194 | 0.129 | 0.190 | 1.000 | 0.329 | 0.329 |
| Forward Stepwise Reg_all | 0.04 | 0.039 | 0.060 | 0.062 | 0.059 | 0.085 | 0.089 | 0.086 | 0.116 | 1.000 | 0.861 | 0.861 |
| LARS_active | 0.073 | 0.054 | 0.094 | 0.104 | 0.078 | 0.131 | 0.118 | 0.081 | 0.134 | 1.000 | 0.382 | 0.382 |
| LARS_all | 0.017 | 0.016 | 0.028 | 0.030 | 0.029 | 0.048 | 0.039 | 0.037 | 0.057 | 1.000 | **0.866** | **0.866** |
| CORTH Features (Ours) | **0.481** | **0.287** | **0.407** | **0.436** | **0.258** | **0.366** | **0.364** | **0.220** | **0.313** | 1.000 | 0.610 | 0.610 |

Table C.9: Performance across all the settings for different number of observation. Each single entry in the table is averaged over 3000 simulations.

| Method | Number of Observations | | | | | | | | | |
|---|---|---|---|---|---|---|---|---|---|---|
| | 10 | | 20 | | 50 | | 100 | | 200 | |
| | CSI | F1 | CSI | F1 | CSI | F1 | CSI | F1 | CSI | F1 |
| Standard Regression | **0.250** | **0.250** | **0.250** | 0.250 | 0.250 | 0.250 | 0.263 | 0.272 | 0.392 | 0.499 |
| Lasso | 0.075 | 0.117 | 0.100 | 0.155 | 0.127 | 0.192 | 0.131 | 0.196 | 0.132 | 0.198 |
| Debiased Lasso | 0.066 | 0.102 | 0.086 | 0.134 | 0.115 | 0.173 | 0.122 | 0.179 | 0.114 | 0.171 |
| Forward Stepwise Reg_active | 0.193 | 0.199 | 0.161 | 0.177 | 0.103 | 0.134 | 0.071 | 0.111 | 0.322 | 0.439 |
| Forward Stepwise Reg_all | 0.222 | 0.224 | 0.222 | 0.226 | 0.229 | 0.236 | 0.244 | 0.257 | 0.389 | 0.458 |
| LARS_active | 0.193 | 0.200 | 0.171 | 0.191 | 0.143 | 0.177 | 0.090 | 0.128 | 0.160 | 0.242 |
| LARS_all | 0.218 | 0.222 | 0.217 | 0.221 | 0.226 | 0.231 | 0.230 | 0.235 | 0.293 | 0.342 |
| CORTH Features (Ours) | 0.125 | 0.173 | **0.250** | **0.314** | **0.353** | **0.445** | **0.443** | **0.548** | **0.550** | **0.640** |

Table C.10: Performance across all the settings for different connectivities. Each single entry in the table is averaged over 5000 simulations.

| Method | Connectivity | | | | | | | | |
|---|---|---|---|---|---|---|---|---|---|
| | 0.1 | | | 0.3 | | | 0.5 | | |
| | TPR | CSI | F1 | TPR | CSI | F1 | TPR | CSI | F1 |
| Standard Regression | 0.395 | 0.267 | 0.290 | 0.372 | 0.292 | 0.313 | 0.350 | 0.284 | 0.310 |
| Lasso | **0.789** | 0.211 | 0.314 | 0.353 | 0.094 | 0.150 | 0.269 | 0.035 | 0.052 |
| Debiased Lasso | 0.787 | 0.211 | 0.314 | 0.296 | 0.054 | 0.087 | 0.270 | 0.037 | 0.055 |
| Forward Stepwise Reg_active | 0.437 | 0.182 | 0.231 | 0.352 | 0.156 | 0.198 | 0.355 | 0.171 | 0.208 |
| Forward Stepwise Reg_all | 0.356 | 0.315 | 0.343 | 0.275 | 0.235 | 0.252 | 0.263 | 0.234 | 0.245 |
| LARS_active | 0.360 | 0.149 | 0.192 | 0.312 | 0.145 | 0.183 | 0.340 | 0.169 | 0.196 |
| LARS_all | 0.289 | 0.246 | 0.264 | 0.267 | 0.233 | 0.246 | 0.259 | 0.231 | 0.239 |
| CORTH Features (Ours) | 0.494 | **0.367** | **0.417** | **0.540** | **0.274** | **0.355** | **0.677** | **0.390** | **0.499** |

Table C.11: Performance across all the settings for different parameters for the Beta distribution. Each single entry in the table is averaged over 3000 simulations.

| Method | Beta Distribution Parameters $(\alpha, \beta)$ | | | | | | | | | |
|---|---|---|---|---|---|---|---|---|---|---|
| | (0.5, 0.5) | | (1, 3) | | (2, 2) | | (2, 5) | | (5, 1) | |
| | CSI | F1 | CSI | F1 | CSI | F1 | CSI | F1 | CSI | F1 |
| Standard Regression | 0.282 | 0.305 | 0.280 | 0.304 | 0.280 | 0.303 | 0.280 | 0.303 | 0.283 | 0.306 |
| Lasso | 0.109 | 0.168 | 0.106 | 0.164 | 0.116 | 0.174 | 0.105 | 0.163 | 0.129 | 0.189 |
| Debiased Lasso | 0.096 | 0.148 | 0.092 | 0.143 | 0.101 | 0.152 | 0.093 | 0.144 | 0.120 | 0.173 |
| Forward Stepwise Reg_active | 0.169 | 0.211 | 0.168 | 0.210 | 0.169 | 0.211 | 0.172 | 0.214 | 0.172 | 0.215 |
| Forward Stepwise Reg_all | 0.261 | 0.279 | 0.257 | 0.275 | 0.265 | 0.284 | 0.259 | 0.278 | 0.266 | 0.284 |
| LARS_active | 0.161 | 0.198 | 0.150 | 0.184 | 0.146 | 0.182 | 0.155 | 0.191 | 0.157 | 0.193 |
| LARS_all | 0.235 | 0.248 | 0.236 | 0.249 | 0.241 | 0.254 | 0.238 | 0.251 | 0.234 | 0.247 |
| CORTH Features (Ours) | **0.336** | **0.417** | **0.330** | **0.411** | **0.354** | **0.434** | **0.336** | **0.417** | **0.363** | **0.441** |

# D    Real-World Data Experiment-Covid19

## D.1    Preprocessing

The preprocessing stage for this dataset is the same as (Schwab et al., 2020) except that, for each target variable upsampling is used to resolve data imbalance.

## D.2    Results

The results obtained by leveraging CORTH Features is suprisingly consistent with (Schwab et al., 2020) which demonstrates the ability of this method in feature selection. The selected features are indicated in Tables D.1 to D.4

Table D.1: Ranks of the features based on the times being predicted as direct causes of **SARS-Cov-2 exam result** out of 1000 different runs of our propsal approach. Not mentiond features were not predicted even once, note that preprocessed dataset has 331 features.

| Rank | Feature | Rate of being Predicted as a Direct Cause |
|---|---|---|
| 1 | Patient age quantile
Arterial Lactic Acid
Promyelocytes
Base excess venous blood gas analysis | 1 |
| 5 | pH venous blood gas analysis | 0.999 |
| 6 | MISSING Mean platelet volume | 0.992 |
| 7 | MISSING Lactic Dehydrogenase | 0.966 |
| 8 | Segmented | 0.934 |
| 9 | Myelocytes | 0.904 |
| 10 | Eosinophils | 0.794 |
| 11 | Leukocytes | 0.784 |
| 12 | Total CO2 arterial blood gas analysis | 0.450 |
| 13 | Potassium | 0.340 |
| 14 | MISSING International normalized ratio INR | 0.289 |
| 15 | Metapneumovirus not detected | 0.234 |
| 16 | Arteiral Fio2 | 0.092 |
| 17 | HCO3 arterial blood gas analysis. | 0.046 |
| 18 | Creatinine | 0.035 |
| 19 | MISSING.Magnesium | 0.034 |
| 20 | pO2 arterial blood gas analysis | 0.031 |
| 21 | MISSING Arteiral Fio2 | 0.024 |
| 22 | Direct Bilirubin | 0.016 |
| 23 | MISSING Ferritin
Respiratory Syncytial Virus detected | 0.014 |
| 25 | MISSING Albumin
Creatine phosphokinase CPK | 0.010 |
| 27 | Strepto A positive | 0.008 |
| 28 | Neutrophils
Red blood cell distribution width RDW
Coronavirus HKU1 detected
Influenza A rapid test positive | 0.004 |
| 32 | Hb saturation venous blood gas analysis | 0.002 |
| 33 | Urine pH
Inf A H1N1 2009 detected
MISSING Serum Glucose
Aspartate transaminase
Urine Esterase nan | 0.001 |

Table D.2: Ranks of the features based on the times being predicted as direct causes of **Patient addmited to regular ward** out of 1000 different runs of our propsal approach. Not mentiond features were not predicted even once, note that preprocessed dataset has 331 features.

| Rank | Feature | Rate of being Predicted as a Direct Cause |
|---|---|---|
| 1 | Patient age quantile
HCO3 venous blood gas analysis
Total CO2 venous blood gas analysis
Gamma glutamyltransferase | 1 |
| 5 | MISSING Lactic Dehydrogenase | 0.987 |
| 6 | Alanine transaminase | 0.845 |
| 7 | MISSING International normalized ratio INR | 0.804 |
| 8 | Serum Glucose | 0.652 |
| 9 | pH venous blood gas analysis | 0.631 |
| 10 | Base.excess venous blood gas analysis | 0.341 |
| 11 | MISSING Arteiral Fio2 | 0.334 |
| 12 | Urine Density | 0.334 |
| 13 | Magnesium | 0.323 |
| 14 | Metapneumovirus not detected | 0.261 |
| 15 | MISSING Mean platelet volume | 0.118 |
| 16 | Creatine phosphokinase CPK | 0.086 |
| 17 | Creatinine | 0.058 |
| 18 | International normalized ratio INR | 0.049 |
| 19 | MISSING Ferritin | 0.046 |
| 20 | Urea | 0.044 |
| 21 | Respiratory Syncytial Virus detected | 0.032 |
| 22 | MISSING Magnesium | 0.021 |
| 23 | MISSING Albumin | 0.018 |
| 24 | MISSING Potassium | 0.016 |
| 25 | Inf A H1N1 2009 detected | 0.014 |
| 26 | Coronavirus HKU1 detected | 0.010 |
| 27 | Strepto A positive | 0.008 |
| 28 | Influenza A rapid test positive | 0.007 |
| 29 | MISSING Sodium
Urine Protein nan | 0.002 |
| 31 | ctO2 arterial blood gas analysis
Influenza A detected
Influenza B detected | 0.001 |

Table D.3: Ranks of the features based on the times being predicted as direct causes of **Patient addmited to semi-intensive unit** out of 1000 different runs of our propsal approach. Not mentiond features were not predicted even once, note that preprocessed dataset has 331 features.

| Rank | Feature | Rate of being Predicted as a Direct Cause |
|---|---|---|
| 1 | Patient age quantile
Creatinine
MISSING Lactic Dehydrogenase
Total CO2 venous blood gas analysis
Magnesium
Gamma glutamyltransferase
Alanine transaminase | 1 |
| 8 | ctO2 arterial blood gas analysis
HCO3 venous blood gas analysis | 0.999 |
| 10 | Relationship Patient Normal | 0.786 |
| 11 | MISSING Arteiral Fio2 | 0.595 |
| 12 | Base excess venous blood gas analysis | 0.578 |
| 13 | pO2 venous blood gas analysis | 0.449 |
| 14 | MISSING International normalized ratio INR | 0.435 |
| 15 | Mean platelet volume | 0.366 |
| 16 | Metapneumovirus not detected | 0.308 |
| 17 | Proteina C reativa mg dL | 0.235 |
| 18 | Sodium | 0.212 |
| 19 | Phosphor | 0.164 |
| 20 | Urine Density | 0.085 |
| 21 | Respiratory Syncytial Virus detected | 0.068 |
| 22 | MISSING Mean platelet volume | 0.056 |
| 23 | MISSING Ferritin | 0.054 |
| 24 | pH venous blood gas analysis | 0.021 |
| 25 | Strepto A positive | 0.018 |
| 26 | Inf A H1N1 2009 detected | 0.016 |
| 27 | Influenza A rapid test positive | 0.014 |
| 28 | MISSING Albumin
Coronavirus HKU1 detected | 0.012 |
| 30 | MISSING Magnesium | 0.008 |
| 31 | Aspartate transaminase | 0.004 |
| 32 | Urine Ketone Bodies absent
Red blood cell distribution width RDW
Influenza A detected
Urine Esterase absent
Urine Protein nan | 0.001 |

Table D.4: Ranks of the features based on the times being predicted as direct causes of **Patient addmited to intensive care unit** out of 1000 different runs of our propsal approach. Not mentiond features were not predicted even once, note that preprocessed dataset has 331 features.

| Rank | Feature | Rate of being Predicted as a Direct Cause |
|---|---|---|
| 1 | Patient age quantile
MISSING Mean platelet volume
Total CO2 venous blood gas analysis
HCO3 venous blood gas analysis
Alanine transaminase
Gamma glutamyltransferase
Magnesium
MISSING Lactic Dehydrogenase
Creatinine | 1 |
| 10 | pO2 venous blood gas analysis | 0.982 |
| 11 | ctO2 arterial blood gas analysis | 0.962 |
| 12 | pH venous blood gas analysis | 0.938 |
| 13 | MISSING Arteiral Fio2 | 0.667 |
| 14 | MISSING International normalized ratio INR | 0.586 |
| 15 | Red blood cell distribution width RDW | 0.503 |
| 16 | Urine Density | 0.414 |
| 17 | Creatine phosphokinase CPK | 0.380 |
| 18 | Base excess venous blood gas analysis | 0.352 |
| 19 | Potassium | 0.234 |
| 20 | Promyelocytes | 0.221 |
| 21 | MISSING Ferritin | 0.174 |
| 22 | Metapneumovirus not detected | 0.132 |
| 23 | Phosphor | 0.082 |
| 24 | Sodium | 0.036 |
| 25 | MISSING Magnesium | 0.032 |
| 26 | Proteina C reativa mg dL | 0.016 |
| 27 | Aspartate transaminase | 0.015 |
| 28 | Respiratory Syncytial Virus detected | 0.010 |
| 29 | Relationship Patient Normal | 0.007 |
| 30 | MISSING Albumin
Arterial Lactic Acid | 0.006 |
| 32 | Coronavirus HKU1 detected
Eosinophils | 0.005 |
| 34 | Inf A H1N1 2009 detected | 0.004 |
| 35 | Influenza A rapid test positive
International normalized ratio INR | 0.002 |
| 37 | Urine Crystals Ausentes
Leukocytes
Strepto A positive | 0.001 |

