# OpenReview forum: "Causal Feature Selection via Orthogonal Search"
_TMLR — Accepted by TMLR_

### Review · Reviewer_Xh2v · 2022-06-03

**Summary Of Contributions:**

In this paper, the authors consider the problem of selecting causal features from an observational dataset. In detail, the authors provide a new method to learn the causal parents of a response variable from the explanatory variables. Besides, the authors show that all parents of the target variable of interest can be identified under certain assumptions. The experimental results demonstrate the effectiveness of their approach.

**Requested Changes:**

Related work: As far as I know, finding the Markov blanket of the response of interest is another line for causal feature selection. Unfortunately, I can not find any descriptions in this paper. One should discuss the connections and differences between their method and  Markov blanket-based methods. In fact, in the paper's setting (there is no hidden confounder for the response variable and all explanatory variables are not the descendants of the response variable), one can identify all parents using the existing Markov blanket-based methods.

The authors may need to compare their method with Markov blanket-based methods in the Experiments section.

﻿"Extensions of the IV-based approach ….are the closest known result to discovering direct causal parents" in Section 1.1: Is this sentence reasonable? Can you explain how?

In the last paragraph on Page 3: no definition for W.

In the first equation on Page 4: no definitions for i and n.

It would be helpful to give the Conclusion Section.

**Strengths And Weaknesses:**

Strengths:

Selecting causal features is an important but challenging problem. The authors extend the method of Double Machine Learning to learn causal parents of the target variable of interest is reasonable.

The analysis and the algorithm are sound and presented in a logical way.


Weaknesses:

The assumptions of linear direct causal effects of the response variable and no hidden confounder for the response variable seem a bit strict.

Lack of some discussion in related work and experiments,  e.g., Markov blanket-based methods.

---

> ### Author Response · Authors · 2022-07-09
> **Reply to Reviewer Xh2v**
>
> Thank you for your feedback on our manuscript. Please find your comments addressed below.
>
> - “The authors may need to compare their method with Markov blanket-based methods”
>
> We added 5 Markov blanket baselines. Markov blanket is not equivalent to our approach to finding direct causes, as we do not require faithfulness. Also Markov blanket approaches combine variables based on some conditional independence measures, and therefore may fail under presence of cycles or hidden confounders between covariates, contrary to our approach. These methods are presented and discussed at multiple places (in red) in the experimental section 3.
>
> - "In the last paragraph on Page 3: no definition for W."
>
> Thank you this is fixed (this is just the collection of all observed variables).
>
> - "In the first equation on Page 4: no definitions for i and n."
>
> Thank you this is now specified (this denotes the indices and number of observed samples).

---

> > ### Comment · Reviewer_Xh2v · 2022-07-16
> > **After Rebuttal**
> >
> > Thanks for your clarification. The authors address my main concerns!

---

### Review · Reviewer_9rrS · 2022-06-06

**Summary Of Contributions:**

The authors aim at the causal feature selection by leveraging the connection with double ML.

1. Assumptions: the model generally assumes that the variable of interest has no children and that the direct causal effects on the variable of interest are linear while the other relationships can be nonlinear and/or contain cycles.
2. Under the assumptions, the authors extend the conclusion in double ML so that it can be applied in more situations that are necessary for the causal feature selection, as in Prop. 2. And then based on the extension, proposed an efficient algorithm as in Alg. 1.
3. Furthermore, a statistical test is provided for determining the direct causal effect, as in Prop. 3.
4. Based on the synthetic data generated according to Eqn (18), the proposed method has state-of-art results. The method is also applied to the COVID-19 dataset.

**Broader Impact Concerns:**

Looks good. I didn't have any concern on the ethical implications of the work.

**Requested Changes:**

**Sec. 2.1**

It seems that the authors assume that the audience has certain knowledge about Neyman's orthogonality condition, i.e., Eqn (3) and (4). However, based on the paper, I don’t know and would like to ask for clarification on several points:

1. how to have the estimator such that $\hat{\eta}$ satisfies Eqn (3).
    * based on the algorithm and the proof for Example 1, it doesn’t seem to require an extra procedure, but only fitting nonlinear regressions.
2. what is the expression of the estimator $\theta$ and how to get it from Eqn. (4).
    * based on the algorithm and the proof for example 1, it seems to have an expression. So I would like to know what is it and how to get it, especially for the audience without sufficient knowledge about Neyman's orthogonality condition and the related works.

**Sec. 2.2**

It would be much better to see the robustness of the proposed algorithm, especially when the assumptions don’t hold. It can be shown via experiments or simply by providing a discussion about them. For example,

1. What if in Z of Eqn. (10), there is a covariate that is not a confounder or a chain. It can be the independent noise or only a parent of Y.
2. What if Y is the cause of some variables.
3. What if there are unknown confounders.

Can the method still work and what would be the consequence?

**Proof of Prop. 2**

- To be self-contained, it would be much more convenient to include Eqn 2.7 and 2.8 in (Chernozhukov et al., 2018a).
- I am not sure how to get the Neyman orthogonal score? It would be better to show the procedures of how to get $\mu$ and $J$. Due to a lack of derivation steps, I didn’t check the correctness of the derivation. Since the derivation is basically plugging in terms, it doesn’t seem to be wrong. But I highly suggest making each step clear for the derivation.
- It would be better to make it clear what is the unbiased estimator expression used in the algorithm. And how to get the expression (maybe the orthogonality condition implies so, but please make it clear).
- The current proof of Prop. 2 is based on linear functions with $\theta$, $\beta$ and $\gamma$. Note that the algorithm and the claims of the paper are based on nonlinear m and g, while the proof of Prop. 2 is based on linear m and g.
    - I am fine with the focus of work is to prove the linear case, but I would suggest making the claims clear, i.e., what has been proved and what hasn't. And why so. In fact, I didn’t see a reason why the nonlinear cases don't hold or are more difficult? I may miss some parts about the nonlinear case; otherwise, I would suggest including the proof of the nonlinear case to make it transparent.
- In the proof of Prop. 2, it assumes the gaussianity of the assignments of Y. It seems that there is no discussion or introduction of this in the main paper. If so, please make the claim and the assumption clear.

**Proof of Prop. 3**

For the proof of Prop. 3, I can understand and follow the proof of a), but I feel it is less straightforward to see how b) is proved and to understand why c) is proved.

**Variance of Empirical Estimates $\chi$**

I would suggest including the important conclusions from the other papers to make them self-contained and easier to read. Moreover, I didn’t see the connection that why after deriving Eqn (11) and the given that Eq (12) is LASSO-type estimators, one can get the asymptotic expression.

**Experiments**

It requires a justification of the data generation process, especially, Eqn. (18).

-----
**Minor comments**
* From page 24 to page 83, they are basically the experiments plots under different configurations without discussion and analysis. It doesn’t seem to be necessary to attach all the experiments plots in this way. I would suggest summarizing them a bit and to show what are the take-away messages. If there are no more things to discuss or analyze, maybe consider keeping the typical ones and removing the others.
* In Figure 1, there are X1 to X6 involved while the caption shows {X1, ..., X11}.
* In Algorithm 1, Line 7, is it supposed to be $\hat{V}_{ij}^{[k]}$?
* In the paragraph after Example 1, it seems that $\theta_2$ fits the double ML model because X1 is a confounder, which can be estimated directly. While the text seems to show that $\theta_1$ is straightforward to be estimated.
* I didn't understand the paragraph above Sec. 2.4.



**Strengths And Weaknesses:**

Strength:
+ In general, the paper is well written. The idea and the method are clearly introduced. For example, I really like the example in Sec. 2.3, which makes the point much easier to be understood.
+ The rigorous analysis and theoretical extension of the double ML are appreciated.
+ The statistical test is appreciated. This is indeed necessary for causal discovery or causal feature selection, which is of lack in current causal methods with ML techniques.

Weakness:

The theoretical analysis and the derivation in Sec. 2.3 and Sec. 2.4 requires more effort to be self-contained and for the clarification and the transparency. The proof and the theoretical results seem right to me, but due to the mentioned weakness, I cannot always follow the proofs and the derivations. It is likely that I may miss or misunderstand some parts, but please clarify the corresponding points if possible to fit a large range of the (potential) readers of TMLR and optimize their benefits of reading the paper. In the following “Requested changes”, I show the detailed comments.

---

### Review · Reviewer_fVm5 · 2022-06-16

**Summary Of Contributions:**

The authors propose a straightforward technique for discovering the causal parents of a target variable, under the assumptions that the structural equation for the target variable is linear in its parents, and all potential parents of the target variable are non-descendants. The technique focuses on each potential parent in turn, and constructs a hypothesis test of whether the conditional covariance (given all other variables) between the potential parent and target is zero, using the double ML framework. The double ML framework uses a Neyman orthogonal score function and cross-fitting to de-bias an estimate of the parameter of interest when nuisance functions are estimated with regularization. The authors construct many simulated DGP's and show that this strategy does a better job of identifying causal parents than more generic causal structure learning algorithms.

**Broader Impact Concerns:**

The real data example concerns COVID diagnosis, which could be a sensitive topic. The paper caveats the inconclusiveness of the analysis relatively well, but if the authors expand on this example, they should remain careful about the strength of claims made here.

**Requested Changes:**

 - Reframe the contributions to focus on debiasing and valid hypothesis tests, rather than identification under arbitrary causal structure in the X's. As stated under Weaknesses, if I understand correctly, the assumptions in the paper imply that ordinary linear regression of Y on all X's would identify the desired causal effects (the controlled direct effect of each X_j on Y), and would thus enable hypothesis testing about whether these effects are zero using more straightforward hypothesis tests, such as those that are built into linear regression routines. These approaches put no conditions on the causal structure of X, but only require that the covariance of X be non-degenerate. It seems that some of the examples attempt to show that the DML approach targets a distinct parameter from what one might target with more naive methods, but this is not the case. For example, the score equation given at the top of page 6 is exactly equivalent to a score equation that is solved by the standard linear regression coefficient on X_k in the multiple regression of Y on all X's, by the Frisch-Waugh-Lovell Theorem. If I am mistaken here, the authors need to provide a better justification of why the double ML procedure targets a unique estimand.

 - (Extension of last point) The more compelling contribution of double ML approaches is that it enables valid hypothesis testing in contexts where nuisance functions need to be estimated with regularization. In linear contexts, this could correspond to cases where the X's are high-dimensional. So it might make sense to re-focus the presentation on high-dimensional contexts.

 - Make sure that the conditions for validity are consistent and complete. As mentioned above, a faithfulness assumption seems to be missing, and assumptions about independence of the noise term are not consistent across the paper.

 - Focus the presentation on the particular hypothesis testing task, rather than the partial linear model.

 - Provide more details of the simulation in the main text. For example, it is very difficult to interpret what the various simulation parameters are from the main text alone.

 - Include baselines from standard linear regression and de-biased LASSO (Javanmard and Montanari).

 - Either eliminate the real data example, or provide a more thorough description of the problem and associated variables in the main text.

**Strengths And Weaknesses:**

Strengths:
 - Overall, the idea of (1) tailoring the solution to the special causal discovery problem of discovering the parents of one variable, and (2) using double ML to construct valid hypothesis tests is nice. This is certainly more efficient than attempting to learn a DAG for all variables, and could work a wide range of cases where nuisance functions needs to be estimated with regularization.
 - The method could scale to high-dimensional cases where regularization like LASSO is necessary to estimate E[Y | X], e.g., the case considered in [Farrell 2015](https://arxiv.org/abs/1309.4686) or [Belloni et al 2014](https://academic.oup.com/restud/article-abstract/81/2/608/1523757?redirectedFrom=fulltext).

Weaknesses:
 - The presentation is far more complex than it needs to be, and does not clearly highlight the advantage of the proposed method. The paper is written as though the Double ML approach targets a different parameter from what other methods would target, but this is not true. The stated assumptions of the problem allow for an immediate reduction of the problem to testing whether the regression coefficient of each X is zero in the linear regression of Y on all X's simultaneously (or, equivalently, whether the partial correlation between Y and each X is zero, after projecting out the rest of the X's). See the [Frisch-Waugh-Lovell Theorem](https://en.wikipedia.org/wiki/Frisch%E2%80%93Waugh%E2%80%93Lovell_theorem) for this equivalence. The causal structure among the X's would not matter for any estimator of E[Y | X], since X is being conditioned on, and the structural noise terms are assumed to be independent of all X's. Rather, the strength of Double ML is providing (potentially) better statistical efficiency and a central limit theorem for hypothesis testing.
 - Some conditions are likely missing. In particular, one needs to have some notion of faithfulness in the X-Y relationship to perform structure discovery, even if the relationships among the X's are irrelevant. Similarly, in the assumptions for the statistical test, equation (9) does not eliminate the case of M-bias, where E[U | X_j] = 0 marginally, but E[U | X_j, X_k] \neq 0, because conditioning on X_k opens a backdoor path. It seems to me that the condition in equation (5) is the one that's actually desired, where E[U | Z, D] = 0.
 - The methodology could be presented more clearly. It seems that the method actually centers around a set of partial covariance tests, not on estimation of parameters in the partial linear model, yet the methodology section begins with a description of estimating treatment effects in the partial linear model. The detour into the partial linear model and treatment effects seems irrelevant.
 - Too many details of the simulations are hidden in the appendix.
 - I am not sure that the baselines being compared to are appropriate. Some straightforward baselines based on hypothesis tests from ordinary linear regression or de-biased LASSO (Javanmard and Montanari) should be added.
 - The real data example does not add any insight, given the level of detail given in the paper.

---

### Decision · Action_Editors · 2022-08-08

**Recommendation:** Accept with minor revision

**Comment:**

This paper studies feature selection for a linear regression SEM when the covariate is in high dimensions. The authors show that under the assumption that no feature in X is a child of Y, they can derive a double ML method (Chernozhukov et al. 2018) to recover the direct parents of Y in X. Computational complexity and asymptotic normality analyses are provided. Upon revision, a variety of experiments have been conducted to demonstrate the improvement of feature selection over other feature selection methods.

Reviewers all recommended "accept/leaning accept" to this submission, so an agreement is reached.

I would ask the authors to again consider the suggestions from reviewer fVm5. Please clarify your assumptions and address fVm5's comment on faithfulness, and add a discussion on comparing your approach with debiased LASSO (not just adding experimental comparisons only). This will help clarify your approach better.